# COMPLEMENTARY CODING OF SPACE WITH COUPLED PLACE CELLS AND GRID CELLS

## ABSTRACT

Spatial coding is a fundamental function of the brain. Place cells in the hippocampus (HPC) and grid cells in the medial entorhinal cortex (MEC) are two primary types of neurons accounting for spatial representation in the brain. These two types of neurons employ different spatial coding strategies and process environmental and motion cues, respectively. In this work, we develop a computational model to elucidate how place and grid cells can complement each other to integrate information optimally and overcome their respective shortcomings. Specifically, we build a model with reciprocally coupled continuous attractor neural networks (CANNs), in which a CANN with location coordinate models the place cell ensemble in HPC, and multiple CANNs with phase coordinate model grid cell modules with different spacings in MEC, and the coupling between place and grid cells conveys the correlation prior between sensory cues. We theoretically derive that the dynamics of our model effectively implements the gradient-based optimization of the posterior. Using simulations, we demonstrate that our model achieves Bayesian optimal integration of the environmental and motion cues, and avoids the non-local error problem in phase coding of grid cells. We hope that this study gives us insights into understanding how place and grid cells complement each other to improve spatial representation in the brain.

## 1 INTRODUCTION

Spatial coding is a fundamental function of the brain, which serves as the foundation for the brain to realize many other advanced cognitive functions. Over decades, experimental studies have unveiled the neural correlates of spatial coding in the brain, and two primary neuron types central to this functionality are identified, which are place cells in the hippocampus (HPC) (O'Keefe & Dostrovsky, 1971; O'Keefe, 1976) and grid cells in the medial entorihnal cortex (MEC) (Hafting et al., 2005). These two types of cells exhibit very different response characteristics in spatial representation. Specifically, place cells display localized place fields, firing intensely at localized locations in the environment; while grid cells display distributed place fields, forming periodic grid-like firing patterns across the environment (Fig.1a). Experimental data further revealed that place and grid cells are primarily responsible for processing different sensory cues, namely, place cells are primarily driven by environmental cues, such as the visual and/or olfactory cues; while grid cells are primarily driven by the self-motion cue of the animal (McNaughton et al., 2006; Laptev & Burgess, 2019; Chen et al., 2019; Sharp et al., 1995).

The significantly different response characteristics between place and grid cells indicate that they employ different strategies to represent space. Simply state, the strategy of HPC is to use intensive number of place cells with localized fields to cover the space compactly (Fig.1a-b), referred to as localized space coding (LSC) hereafter; while the strategy of MEC is to use the combination of grid cells' phases with varying spacing to encode space (Fig.1c-d), referred to as phase space coding (PSC) hereafter. From the information theory point of view, LSC is robust to noise but not efficient, since the number of neurons it needs increases linearly with the spatial size (Abbott & Dayan, 1999; Latham et al., 2003); while PSC is efficient but susceptible to noise, as small perturbations in grid cells' activities can induce non-local errors in the space representation (Fiete et al., 2008b; Sreenivasan & Fiete, 2011) (Fig.1c).

Given that place and grid cells employ different coding strategies and process different sensory cues, a prompt question is: whether these two coding strategies can complement with each other, such that the brain can integrate two sensory cues optimally to improve spatial representation? Experimental studies have shown that there exist abundant reciprocal connections between HPC and MEC (Manns & Eichenbaum, 2006; Wible, 2013; Bush et al., 2014), which can support the interaction between place and grid cells. In this work, we build a computational model with reciprocally coupled place and grid cells, and demonstrate that HPC and MEC can interact with each other to integrate the environmental and motion cues optimally for spatial representation.

The organization of the paper are as follows. In Sec.2, we first review the different coding properties of place and grid cells, and discuss about their respective advantages and disadvantages in spatial representation. In Sec.3, we use a probabilistic inference model to elucidate the information integration between place and grid cells, and present an optimal decoding approach called gradient-based optimization of posterior (GOP). In Sec.4, we build a computational model with reciprocally coupled continuous attractor neural networks (CANNs), in which a CANN with location coordinate models the place cell ensemble in HPC, and multiple CANNs with phase coordinate model the grid cell modules of different spacings in MEC. HPC and MEC receive the location information through the environmental and the motion cues, respectively, and they are reciprocally connected in a congruent manner, which conveys the correlation prior between two sensory cues. We theoretically derive that our model implements GOP effectively. In Sec.5, we carry out simulations to demonstrate that our model integrates two sensory cues optimally in a Bayesian optimal manner, and that it avoids the non-local error problem of PSC. In Sec.6, overall conclusions of this study and related works are discussed.

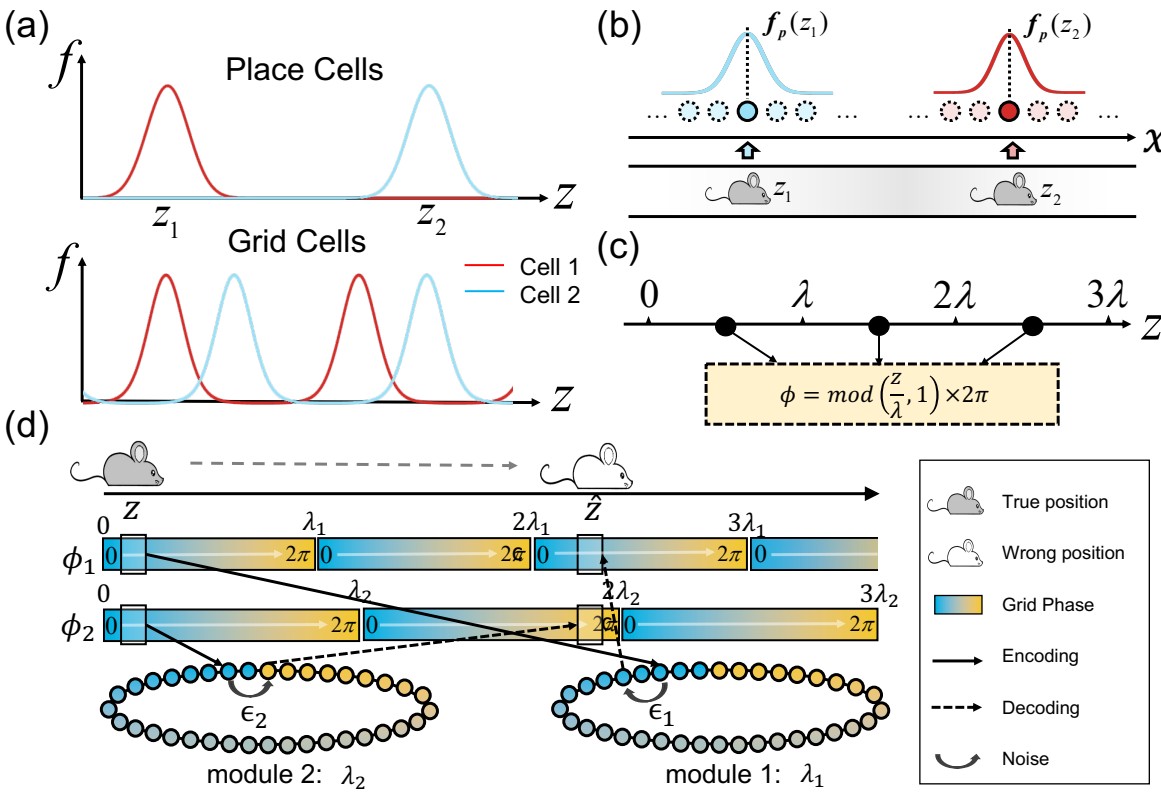

Figure 1: Different coding strategies of place and grid cells. **a.** Tuning curves of place cells (up) and grid cells (down). A place cell fires at a localized position. A grid cell fires at periodic positions, which are multiple locations equally spaced in the one-dimensional case. **b.** Illustration of localized space coding (LSC) of place cells. A bump-shaped population activity of the place cell ensemble encodes an animal location in the space. **c.** Illustration of phase space coding (PSC) of grid cells. Multiple positions with spacing $\lambda$ are represented by a single phase value $\phi$, which mathematically corresponds to a remainder operation. **d.** Illustration of the non-local error of PSC. An animal location $z$ is represented by the combination of two phase values $\phi_1$ and $\phi_2$ in two grid cell modules with spacing $\lambda_1$ and $\lambda_2$, respectively. Small fluctuations in phases, $\phi_1 + \epsilon_1$ and $\phi_2 + \epsilon_2$, can induce a non-local error in the space representation, resulting from $z$ to $\hat{z}$.

## 2    DIFFERENT CODING STRATEGIES OF PLACE CELLS AND GRID CELLS

We start to review the different properties of place and grid cells for spatial representation. In the present study, for the simplicity of analysis, we focus on the encoding of one-dimensional (1D) space, which corresponds to linear tracks widely used in experiments (e.g., (Skaggs et al., 1996)). The main conclusions of this work are, however, extendable to the 2D space.

Place cells perceive the animal location primarily through the environmental cue, e.g., the visual and/or olfactory cues. Let us denote $z \in (-L/2, L/2)$ the animal location, with $L$ the spatial range. The response of a place cell

with preferred location $x$ can be written as,

$$r_p(x; z) = f_p(x; z) + \sigma_p \epsilon_p(x) = A_p \exp\left[-\frac{(x-z)^2}{4a_p^2}\right] + \sigma_p \epsilon_p(x), \tag{1}$$

where $f_p(x; z)$ is the tuning curve, $A_p$ and $a_p$ the peak and width of the tuning function, respectively. $\epsilon_p(x)$ denotes Gaussian noise of zero mean and unit variance, and $\sigma_p$ the noise strength. The joint activity of place cells, denoted as $\boldsymbol{r}_p(z) = \{r_p(x; z)\}$, for $x \in (-L/2, L/2)$, encodes the animal location $z$ as a localized activity bump in the space, which forms LSC (see Fig.1b).

Grid cells perceive the animal location primarily through the self-motion cue, and a location $z$ is expressed as phase $\phi \in (0, 2\pi)$ of a grid cell module according to the conversion rule, $\phi(z) = \mod(z/\lambda, 1) \times 2\pi$, where $\lambda$ denotes the periodicity of the place field, called spacing, and the symbol $\mod(\cdot)$ denotes the remainder operation (Fig.1c). In a single module, a phase value corresponds to multiple spatial locations as induced by the remainder operation, implying that a grid cell fires at multiple locations. This gives rise to the place field of the hexagon-shape in the 2D space and equally spaced firing positions in the 1D space (Fig.1a,c). To overcome the representation ambiguity of a single module, MEC includes multiple modules with varying spacings, and uses the combination of grid cells' phases from different modules to encode a position, denoted as $\boldsymbol{\phi}(z) = \{\phi^i(z)\}$, for $i = 1, \ldots, M$, with $\phi^i(z) = \mod(z/\lambda_i, 1) \times 2\pi$ and $\lambda_i$ the spacing of the $i$th module (Fig.1d). Given that $\{\lambda_i\}$ are prime numbers, the spatial range represented by MEC unambiguously is $\prod_i \lambda_i$.

Let us denote $\phi^i(z)$ the phase of the $i$th module corresponding to the animal location $z$. The response of a grid cell in the $i$th module with preferred phase at $\theta^i$ is written as,

$$r_g(\theta^i; \phi^i) = f_g(\theta^i; \phi^i) + \sigma_{gi}\epsilon_g(\theta^i) = A_g \exp\left\{-\frac{||\theta^i - \phi^i||^2}{4a_{gi}^2}\right\} + \sigma_{gi}\epsilon_g(\theta^i), \tag{2}$$

where $f_g(\theta^i; \phi^i)$ is the tuning curve, $A_g$ and $a_{gi}$ the peak and width of the tuning curve, respectively. $\epsilon_g(\theta^i)$ represents Gaussian noise of zero mean and unit variance, and $\sigma_{gi}$ the noise strength. The symbol $||\theta^i - \phi^i|| = \min\left(|\theta^i - \phi^i|, 2\pi - |\theta^i - \phi^i|\right)$, reflecting the periodicity of phase.

## 2.1 CODING EFFICIENCY

The efficiency of a coding strategy can be quantified by information rate, which is defined a $R = \ln L/N$, with $L$ the spatial range and $N$ the number of neurons (Mackay, 2003; Sreenivasan & Fiete, 2011). We can roughly estimate the efficiency of LSC and PSC as follows. In LSC, the place cell ensemble covers the space compactly. Denote $\Delta x$ to be the interval covered by a place cell with its local place field. The whole space range covered by $N$ place cells is $L = N\Delta x$. Thus, the information rate of LSC is given by $R_p = \log(N\Delta x)/N$, which becomes very small for large $N$, indicating that LSC is not efficient, in term of utilizing the resource of neurons. In PSC, the phases of grid cells in a single module cover a spatial range $\lambda$ (the spacing); beyond that, multiple positions share the same phase value, inducing ambiguity. By employing multiple modules of varying spacing $\lambda_i$, for $i = 1, \ldots, M$, and that $\{\lambda_i\}$ are prime numbers, the spatial range unambiguously represented by the combination of grid cells' phases is $L = \prod_{i=1}^{M} \lambda_i \approx \bar{\lambda}^M$ (Fiete et al., 2008a). Thus, the information rate of PSC is given by $R_g = \log(\bar{\lambda}^M)/MN_0 = \log \bar{\lambda}/N_0$, with $N_0$ the number of neurons in each module. This shows that with the increased number of modules $M$ (and so does the number of neurons $MN_0$), the information rate of PSC is a constant, indicating that PSC is much more efficient than LSC in utilizing the resource of neurons (for more detailed analysis, see Appendix.A).

## 2.2 CODING ACCURACY

We use Fisher information, whose inverse is the lower bound of decoding variance of any unbiased estimator (commonly known as the Cramér–Rao bound (CRAMÉR, 1946)), to quantify the decoding accuracy of LSC. For the response property of place cells given by Eq. (1), the Fisher information can be analytically calculated, whose value increases linearly with the number of neurons, indicating that LSC can achieve arbitrary accuracy if the number of neurons is sufficiently large (see Appendix.A). The accuracy of PSC is, however, very sensitive to noises. In PSC, grid cells' phases are combined to represent spatial location. Due to the periodic mapping between phase and location, two points close in the phase plane can represent two far away spatial locations (see illustration in Fig.1d and details in Appendix.A). This implies that small fluctuations in grid cells' activities can induce large non-local errors in spatial representation, a shortcoming of PSC that has been noticed previously (Sreenivasan & Fiete, 2011).

# 3 A PROBABILISTIC MODEL OF INFORMATION INTEGRATION BETWEEN PLACE AND GRID CELLS

Given that place and grid cells implement different coding strategies and that they process different sensory cues, a nature question is: how can they complement with each other to improve spatial representation? Before going to the detail of neural implementation, we first introduce a probabilistic inference model which elucidates theoretically how the information from two types of cells can be integrated (see Fig. 2a). For the convenience of analysis, we consider that noises in neuronal responses are all independent to each other.

## 3.1 THE ENCODING PROCESS

The brain perceives the animal location via both environment and motion cues, and the two cues go through physically separate pathways. This implies that the information about the animal location conveyed to place and grid cells are highly correlated but not exactly the same (because of the different reliability of the sensory cues and the independent noises in signal transmission). To reflect this correlation relationship, we set the joint property $p(z, \phi)$ of the animal location $z$ perceived by place cells and the phases $\phi$ perceived grid cells satisfy,

$$p(z, \phi) = \prod_{i=1}^{M} p(z, \phi^i) = \prod_{i=1}^{M} \frac{1}{\sqrt{2\pi}\sigma_{\phi^i}} \exp\left\{-\frac{||\phi^i - \psi^i(z)||^2}{2\sigma_{\phi^i}^2}\right\}, \tag{3}$$

where $\psi^i(z)$ is the phase value matching the location $z$, i.e., $\psi^i(z) = \mod(z/\lambda_i, 1) \times 2\pi$, and $\sigma_{\phi^i}$ controls the correlation level between two sensory cues. In the above, we have used the condition that grid cell modules are independent to each other as a consequence of no connection between them, which gives $p(z, \phi) = \prod_{i=1}^{M} p(z, \phi^i)$. This form of joint probability implies that the locations represented by place cells and grid cells tend to be close, but may be different because of noises. This joint probability serves as the correlation prior for integrating information between place and grid cells (see below).

According to the response property of Eq. (1), the likelihood function of observing the joint activity $r_p$ of place cells given the animal location $z$ is written as,

$$p(r_p|z) = \prod_x p\left[r_p(x)|z\right] = \prod_x \frac{1}{\sqrt{2\pi}\sigma_p} \exp\left\{-\frac{[r_p(x) - f_p(x; z)]^2}{2\sigma_p^2}\right\}. \tag{4}$$

According to the response property of Eq. (2), the likelihood function of observing the joint activity $r_g$ of grid cells given the phases $\phi$ is written as,

$$p(r_g|\phi) = \prod_{i=1}^{M} \prod_{\theta^i} p\left[r_g(\theta^i)|\phi^i\right] = \prod_{i=1}^{M} \prod_{\theta^i} \frac{1}{\sqrt{2\pi}\sigma_g} \exp\left\{-\frac{\left[r_g(\theta^i) - f_g(\theta^i; \phi^i)\right]^2}{2\sigma_g^2}\right\}. \tag{5}$$

## 3.2 THE DECODING PROCESS

Given the observed responses $(r_p, r_g)$ of place and grid cells, the neural decoder infers the animal location $z$ represented by place cells and the phases $\phi$ represented by grid cells. According to the Bayes' theorem, the posterior distribution of $(z, \phi)$ given $(r_p, r_g)$ is written as

$$p(z, \phi|r_g, r_p) \propto p(r_p, r_g|z, \phi)p(z, \phi) = p(r_p|z)p(r_g|\phi)p(z, \phi)$$

$$= \prod_x p\left[r_p(x)|z\right] \prod_{i=1}^{M} \left\{ p(z, \phi^i) \prod_{\theta^i} p\left[r_g(\theta^i)|\phi^i\right] \right\}. \tag{6}$$

In the above, we have used the condition that the environmental and motion cues go through separate pathways, which gives $p(r_p, r_g|z, \phi) = p(r_p|z)p(r_g|\phi)$.

Maximum a posteriori (MAP) decoding is a widely used method and can be proven to be the optimal decoding strategy under a 0-1 loss function. However, in practice, the high non-linearity of the posterior often makes direct maximization computationally challenging, necessitating the use of approximation methods. The gradient-based optimization of posterior (GOP) is such an approximation approach (Wu et al., 2017). It starts from an initial point to search for a solution by ascending along the gradient of the log posterior until a local maximum is reached. By this, GOP finds an approximated maximum of the posterior. Interestingly, we find that in the case of phase coding, GOP tends to outperform MAP, as it avoids large non-local errors (see below).

Combining Eqs. (3-6), the gradients of the posterior are calculated to be (for details, see Appendix.B),

$$\frac{\partial}{\partial z} \ln p(z, \boldsymbol{\phi} | \boldsymbol{r}_g, \boldsymbol{r}_p) = \sum_i \frac{2\pi}{\lambda_i \sigma_{\phi^i}^2} ||\phi^i - \psi^i(z)|| + \frac{\rho_p}{\sigma_p^2} \int r_p(x) \frac{\partial f_p(x; z)}{\partial z} dx, \tag{7}$$

$$\frac{\partial}{\partial \phi^i} \ln p(z, \boldsymbol{\phi} | \boldsymbol{r}_g, \boldsymbol{r}_p) = \frac{1}{\sigma_{\phi^i}^2} ||\psi^i(z) - \phi^i|| + \frac{\rho_g}{\sigma_{gi}^2} \int r_g(\theta^i) \frac{\partial f_g(\theta^i; \phi^i)}{\partial \phi^i} d\theta^i, \quad i = 1, \dots, M. \tag{8}$$

In the below, we will show that a biologically plausible network model can implement GOP and that GOP overcomes the non-local error of PSC.

(a)

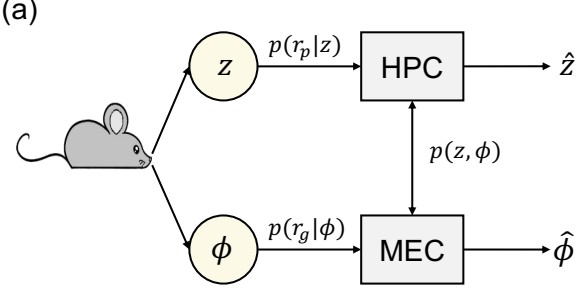

$p(r_p|z)$: likelihood function corresponding to the environmental cue.

$p(r_g|\phi)$: likelihood function corresponding to the motion cue.

$p(z, \phi)$: Correlation prior between the environmental and the motion cue.

(b)

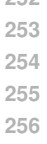
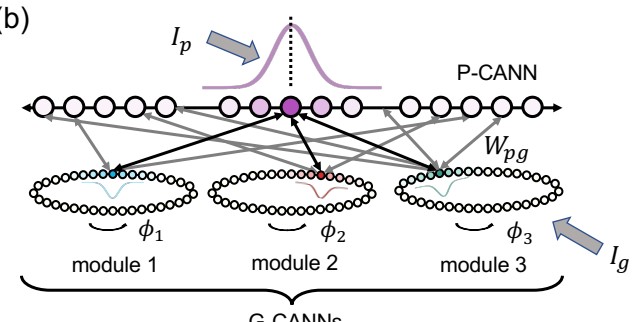

$I_p$: external inputs to place cells (the environmental cue)

$I_g$: external inputs to grid cells (the motion cue)

$W_{pg}$: Reciprocal connection between place cells and grid cells, conveying the correlation prior.

Figure 2: Complementary coding of space with place and grid cells. **a**. A probabilistic inference model describing the information integration between place and grid cells. The animal location $z$ is encoded by the population activity $\boldsymbol{r}_p$ of place cells in HPC in the form of the likelihood function $p(\boldsymbol{r}_p|z)$ and by the population activity $\boldsymbol{r}_g$ of grid cells in MEC in the form of the likelihood function $p(\boldsymbol{r}_p|\boldsymbol{\phi})$. By integrating two sensory cues with the correlation prior $p(z, \boldsymbol{\phi})$, HPC and MEC output the animal location $\hat{z}$ and the corresponding phases $\hat{\boldsymbol{\phi}}$, respectively. **b**. A network model with coupled place and grid cells implementing information integration between sensory cues. The place cell ensemble is modeled as a 1D CANN with location coordinate (P-CANN). A grid cell module is modeled as a 1D CANN with phase coordinate (G-CANN), and three moduels are illustrated. The P-CANN and each G-CANN are reciprocally connected, and no connection between G-CANNs exists.

## 4 INFORMATION INTEGRATION WITH RECIPROCALLY CONNECTED PLACE AND GRID CELLS

In this section, we develop a biologically plausible network model to implement information integration between place and grid cells as described in Sec.3.

### 4.1 A NETWORK MODEL WITH COUPLED PLACE AND GRID CELLS

As shown in Fig.2b, we use a 1D CANN with location coordinate to model the place cell ensemble in HPC, referred to as P-CANN, in which neurons are aligned according to their preferred locations $x \in (-L/2, L/2)$ in the 1D space. We use multiple 1D CANNs with phase coordinate to model grid cell modules with different spacings in MEC, referred to as G-CANNs, and in each G-CANN, neurons are aligned according to their preferred phases $\theta \in (0, 2\pi]$. The P-CANN and each G-CANN are reciprocally connected, and no connection between G-CANNs exists.

The recurrent connections between neurons in the same network are set as follows. For two neurons at locations $x$ and $x'$ in the P-CANN, their connection is $W_p(x, x') = J_p/(\sqrt{2\pi}a_p) \exp\left[-(x - x')^2/2a_p^2\right]$, with $a_p$ controlling the range of recurrent connections and $J_p$ the connection strength. For two neurons at phases $\theta$ and $\theta'$ in a G-CANN, their connection is $W_g(\theta, \theta') = J_g/(\sqrt{2\pi}a_{gi}) \exp\left[-||\theta - \theta'||^2/2a_{gi}^2\right]$. To realize the correlation prior Eq. (3), we set reciprocal connections between place cells at location $x$ and grid cells at phase $\theta^i$ as,

$$W_{gi,p}(x, \theta^i) = \frac{J_{gi,p}}{\sqrt{2\pi}a_{gi,p}} \exp\left[-\frac{||\theta^i - \psi^i(x)||^2}{2a_{gi,p}^2}\right], \tag{9}$$

where $\psi^i(x) = \mathrm{mod}(x/\lambda^i, 1) \times 2\pi$ is the phase value matching the location $x$. This connection form is called congruent, in term of that place and grid cells tend to have strong connections when their preferred location and phase are matched.

Denote $U_p(x, t)$ the synaptic current to place cells at location $x$ at time $t$, and $R_p(x, t)$ the corresponding firing rate. The dynamics of place cells are written as,

$$\tau_p \frac{dU_p(x, t)}{dt} = -U_p(x, t) + \rho_p \int_{-L/2}^{L/2} W_p(x, x')R_p(x', t)dx'$$
$$+ \sum_{i=1}^{M} \rho_g \int_{-\pi}^{\pi} W_{gi,p}(x, \theta^i)R_g(\theta^i, t)d\theta^i + I_p(x), \tag{10}$$

where $\tau_p$ denotes the time constant, $\rho_p$ and $\rho_g$ the densities of place and grid cells, respectively. $I_p(x)$ represents the external inputs conveying the location information from the environmental cue.

Denote $U_g(\theta^i, t)$ the synaptic current at time $t$ to grid cells at phase $\theta^i$ in the $i$th module, and $R_g(\theta, t)$ the corresponding firing rate. The dynamics of grid cells are written as,

$$\tau_g \frac{dU_g(\theta^i, t)}{dt} = -U_g(\theta^i, t) + \rho_g \int_{-\pi}^{\pi} W_g(\theta^i, \theta'^i)R_g(\theta'^i, t)d\theta'^i$$
$$+ \rho_p \int_{-L/2}^{L/2} W_{gi,p}(x, \theta^i)R_p(x, t)dx + I_g(\theta^i), \quad i = 1, \ldots, M, \tag{11}$$

where $\tau_g$ denotes the time constant, and $I_g(\theta^i)$ the external inputs conveying the location information from the motion cue.

The external inputs to the networks convey the location information of the animal. We set them to be,

$$I_p(x) = \alpha_p \left\{\exp\left[-\frac{(x - z_0)^2}{4a_p^2}\right] + \xi_p\right\}, \tag{12}$$

$$I_g(\theta^i) = \alpha_{gi} \left\{\exp\left[-\frac{||\theta^i - \psi^i(z_0)||^2}{4a_{gi}^2}\right] + \xi_g\right\}, \quad i = 1, \ldots, M, \tag{13}$$

where $z_0$ denotes the true location of the animal and and $\psi^i(z_0)$ the matched phase. $\alpha_{gi}$ and $\alpha_p$ represent the strengths of the external inputs. $\xi_g$ and $\xi_p$ are Gaussian noises of zero mean and variances $\sigma_{gi}^2$ and $\sigma_p^2$, reflecting the ambiguity of sensory cues.

For both place and grid cells, the relationships between neuronal synaptic current and firing rate are expressed as,

$$R_p(x, t) = \frac{U_p(x, t)^2}{1 + k_p\rho_p \int U(x', t)^2 dx'}, \quad R_g(\theta^i, t) = \frac{U_g(\theta^i, t)^2}{1 + k_g\rho_g \int U_g(\theta'^i, t)^2 d\theta'^i}, \tag{14}$$

where $k_p$ and $k_g$ control the amplitude of divisive normalization (Hao et al., 2009).

When no reciprocal connection between HPC and MEC ($J_{p,g} = 0$) and no external synaptic input ($I_p = I_g = 0$) exists, the P-CANN or each G-CANN holds a continuous family of Gaussian-shaped stationary states due to the property of CANNs (Fung et al., 2012; 2010; Wong et al., 2008), and they are expressed as,

$$\bar{U}_p(x) = A_p \exp\left[-\frac{(x - z)^2}{4a_p^2}\right], \quad \bar{U}_g(\theta^i) = A_g \exp\left[-\frac{||\theta^i - \phi^i||^2}{4a_{gi}^2}\right], \tag{15}$$

where $z$ and $\phi^i$ are free parameters, $A_p(A_g)$ and $a_p(a_{gi})$ denote, respectively, the height and width of the activity bump of place (grid) cells (see details in Appendix.C).

## 4.2 THE MODEL IMPLEMENTING GRADIENT-BASED OPTIMIZATION OF POSTERIOR

We analyze the model dynamics theoretically. Previous studies have shown that a key property of a CANN is that its dynamics is dominated by very few motion modes, and we can project the CANN dynamics onto these dominating modes to simplify the network dynamics significantly (Fung et al., 2010; Wong et al., 2008) (projecting a function $f(x,t)$ onto a motion mode $u(t)$ means computing $\int_x f(x,t)u(x)dx$). Here, we consider the first two dominating motion modes representing the height and position variations of the bump, respectively, which are,

$$u_T^0(s) = \bar{U}_T(s), \quad u_T^1(s) = \frac{\partial \bar{U}_T(s)}{\partial s}, \quad \text{for } T = p, g; s = x, \theta^i, \tag{16}$$

where $\bar{U}_T(s)$ is the stationary state of P-CANN ($T = p, s = x$) or G-CANN ($T = g, s = \theta^i$) as given in Eq. (15).

For coupled CANNs, we assume that the states of both P-CANN and G-CANN can still be well approximated as of the bump-shape given by Eq. (15), which is confirmed by our numerical simulations (see Appendix.D). Substituting Eq. (15) into the model dynamics Eqs. (10,11) and then projecting them onto the two dominating modes Eq. (16), we obtain the simplified model dynamics, which are the dynamics of the bump center of place cells $z(t)$ and the bump centers of grid cells $\phi^i(t)$, for $i = 1, \ldots, M$. They are written as (see Appendix.D for details):

$$\frac{dz}{dt} = \frac{1}{\tau_p A_p}\left\{\frac{1}{4}\sum_i \frac{\lambda_i}{2\pi}J_{g,p}\rho_g\hat{R}_g||\phi^i - \psi^i(z)|| + \frac{1}{2\sqrt{\pi}a_p}\int I_p(x)u_p^1(x)dx\right\}, \tag{17}$$

$$\frac{d\phi^i}{dt} = \frac{1}{\tau_g A_g}\left\{\frac{\lambda}{8\pi}J_{g,p}\rho_p\hat{R}_p||\psi^i(z) - \phi^i|| + \frac{1}{2\sqrt{\pi}a_{gi}}\int I_g(\theta^i)u_g^1(\theta^i)d\theta^i\right\}, i = 1, \ldots, M, \tag{18}$$

where $\hat{R}_p$ and $\hat{R}_g$ represent the maximum firing rates of place cells and grid cells, respectively. The first terms on the right sides of Eqs.(17,18) are the contributions of reciprocal interactions between place and grid cells, and the second terms are the contributions of the external inputs.

It can be checked that when the parameters satisfy the condition, $2\pi A_p\tau_p\rho_p R_p = \lambda_i A_g^i\tau_g\rho_g^i R_g^i$, the above dynamics can be reorganized as (Appendix.D),

$$\frac{dz}{dt} = \frac{\partial}{\partial z}\ln p(z, \boldsymbol{\phi}|\boldsymbol{r}_p, \boldsymbol{r}_g), \quad \frac{d\phi^i}{dt} = \frac{\partial}{\partial \phi^i}\ln p(z, \boldsymbol{\phi}|\boldsymbol{r}_p, \boldsymbol{r}_g), \tag{19}$$

where the variance terms in $\ln p(z, \boldsymbol{\phi}|\boldsymbol{r}_p, \boldsymbol{r}_g)$ are given by $\sigma_p^2 = (\sqrt{\pi}A_p^3\rho_p\tau_p)/(a_p\alpha_p)$, $\sigma_{gi}^2 = (\sqrt{\pi}A_g^3\rho_g\tau_g)/(a_{gi}\alpha_{gi})$, and $\sigma_{\phi^i}^2 = (8\pi A_g\tau_g)/(\lambda_i J_{gi,p}\rho_p\hat{R}_p)$.

Note that $\sigma_{\phi^i}$ specifies the correlation level between sensory cues (Eq. 3) and $J_{gi,p}$ specifies the reciprocal connection strength between place and grid cells (Eq. 9). The relationship $\sigma_{\phi^i}^2 = (8\pi A_g\tau_g)/(\lambda_i J_{gi,p}\rho_p\hat{R}_p)$ indicates that the reciprocal connections between HPC and MEC conveys the correlation prior between environment and motion cues on animal location.

Finally, comparing Eq. (19) with (7,8), we see that the model dynamics effectively implements GOP, i.e., the Bayesian integration of information.

It is interesting to note that the simplified model dynamics Eq. (19) have a Lyapunov function, which is the negative posterior $L = -\ln p(z, \boldsymbol{\phi}|\boldsymbol{r}_p, \boldsymbol{r}_g)$, whose dynamics is given by

$$\frac{dL}{dt} = -\frac{\partial(\ln p)}{\partial \boldsymbol{y}}\frac{d\boldsymbol{y}}{dt} = -\left(\frac{d\boldsymbol{y}}{dt}\right)^2 \leq 0, \quad \text{for } \boldsymbol{y} = \{z, \boldsymbol{\phi}\}. \tag{20}$$

This ensures that the model dynamics always converge to a local maximum of the posterior.

## 5 SIMULATION EXPERIMENTS

We carry out simulation experiments to further confirm the above theoretical analysis.

### 5.1 THE MODEL IMPLEMENTING BAYESIAN INTEGRATION OF INFORMATION

We first demonstrate that our network model implements GOP as theoretically analyzed. We consider a scenario where the animal is positioned at a fixed location ($x = 0$) on a linear track ranging from $-30$ to $30$, with the

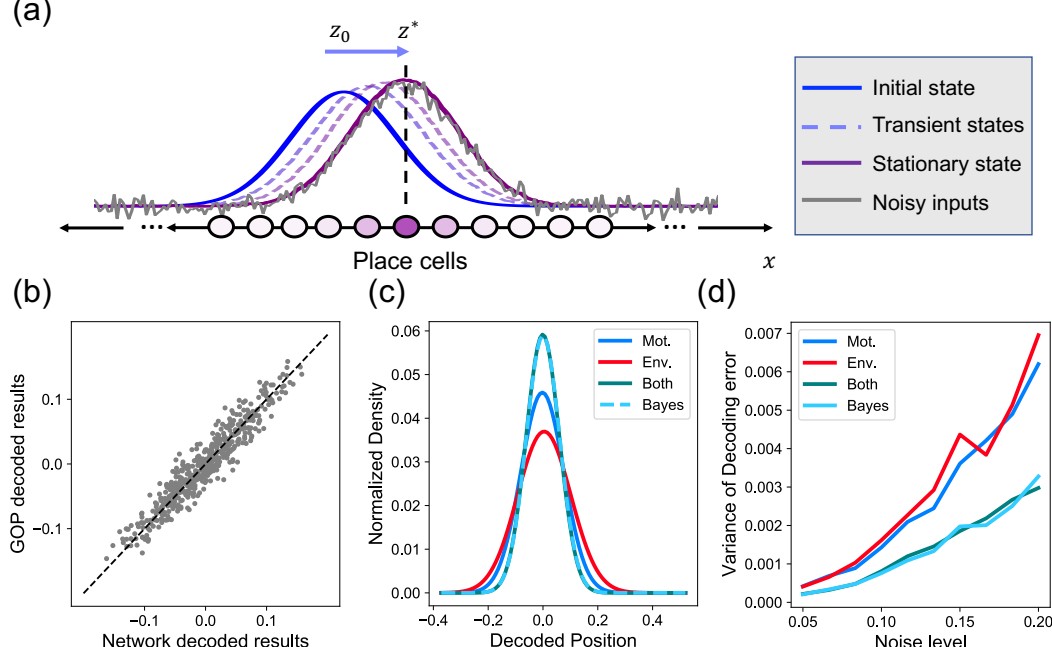

Figure 3: Optimal information integration by the coupled CANNs. **(a)**. Illustration of the network decoding process. The bump activity of place cells is initially at $z_0$ (blue line). In response to noisy inputs (gray line), the bump moves to a stationary state (purple line), whose position $z^*$ outputs the decoding result. **(b)**. Comparing the model decoding results and that of GOP. Each dot (gray) represents the result of single trial. **(c)**. Fitting the distributions of decoding results with Gaussian functions. The blue, red, and green lines correspond to the results of (1) only presenting the motion cue (Mot.), (2) only presenting the environmental cue (Env.), and (3) presenting both cues (Both). The dashed blue line represents the prediction of Bayesian integration (Bayes) based on the results of (1) and (2). Note the the network decoding results almost completely overlap with the Bayesian prediction. **(d)**. Comparing the decoding variances in three cuing conditions and that of Bayesian prediction across different noise levels. For details of the simulation experiments, see Appendix D.

network receiving two noisy inputs representing environmental and motion cues, respectively. These noisy inputs are generated according to Eqs. (12,13).

We apply two methods to decode the animal location. One is to apply GOP to directly calculate the local maximum of the posterior according to Eqs.(7-8) (see Appendix B). The other to run the network dynamics given by Eqs.(10-11) and read out the animal location based on the bump position ($z$) of place cells (see illustration in Fig 3a). The decoding results of two methods are shown in Fig 3b. We see that they agree with each other very well, confirming that the network model implements GOP effectively.

We further demonstrate that our network model performs optimal cue integration. To this end, we consider three cuing conditions: 1) only the environmental cue is presented, corresponding to only place cells receiving external inputs ($I_p \neq 0$ and $I_g = 0$); 2) only the motion cue is presented, corresponding to only grid cells receiving external inputs ($I_p = 0$ and $I_g \neq 0$); 3) both the environmental and motion cues are presented, corresponding to both place and grid cells receiving external inputs ($I_p \neq 0$ and $I_g \neq 0$). In each condition, we run the network dynamics and read out the bump position ($z$) of place cells as the neural representation of the animal location. For comparison, we also calculate the theoretical prediction of Bayesian cue integration. Under the assumptions of Gaussian distributions of decoding errors and the uniform prior, the variance of Bayesian cue integration is given by $1/\sigma_{\text{bayes}}^2 = 1/\sigma_{\text{env}}^2 + 1/\sigma_{\text{mot}}^2$, where $\sigma_{\text{env}}^2$ and $\sigma_{\text{mot}}^2$ represent the variances of decoding errors of using only the environmental or the only motion cue, respectively (Ernst & Banks, 2002). We also vary the noise levels of external inputs ($\sigma_p^2$ and $\sigma_g^2$ in Eqs.(12) and (13)) to assess the robustness of model integration. Fig.3c-d present the simulation results, which show that across different noise levels, the model integration consistently yields lower decoding errors compared to that using a single cue, and that the model decoding results matches well with the theoretical prediction of Bayesian integration, confirming that our model integrates sensory cues optimally. For details of the experiments and calculations, see Appendix D.

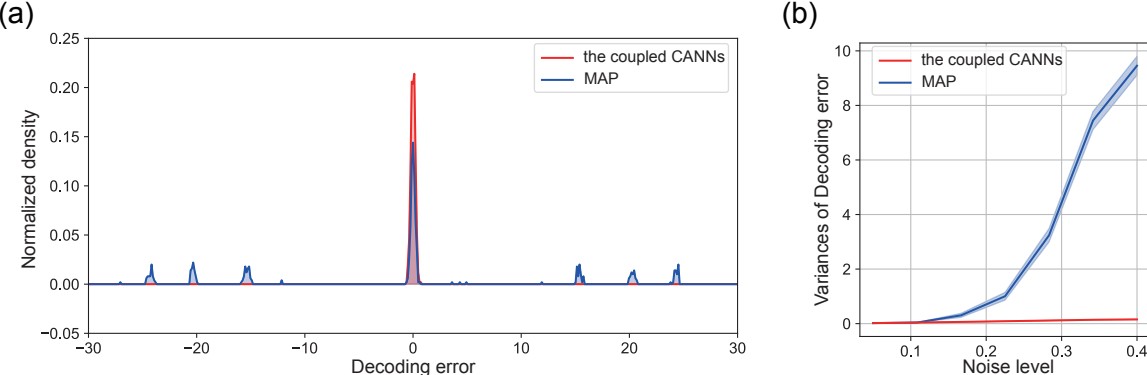

Figure 4: The coupled CANNs resolves the non-local error in phase coding of grid cells. **(a)**. An example of the distributions of decoding errors using the coupled CANNs (red line) and that of maximum a posteriori (MAP) based on the grid cell activity only (blue line). The decoding errors of the network model concentrate around the true animal location, while the decoding errors of MAP distribute widely. **(b)**. Decoding errors across different noise levels using the coupled CANNs (red line) and MAP based on the grid cell activity only (blue line). For details of the simulation experiments, see Appendix D.

### 5.2 THE MODEL ELIMINATING NON-LOCAL ERRORS

In certain scenarios, such as moving in dark, animals have to locate themselves relying mainly on the motion cue. In our model, this corresponds to the external inputs $I_p = 0$ and $I_g \neq 0$. In such a case, if we only use the responses of grid cells to decode the position, e.g., by maximizing $p(r_g|\phi)$ directly, we will get large non-local errors associated with phase coding (Sreenivasan & Fiete, 2011). On the other hand, if we let the coupled CANNs to decode the location, which is given by the bump position of place cells, non-local errors are eliminated, as shown in Fig. 4a. The underlying reason is intuitively understandable. In the coupled CANNs, although place cells does not receive the environment cue ($I_p = 0$), through reciprocal connections between networks, place cells will still generate a bump activity to represent the animal location. As analyzed above, the dynamics of coupled CANNs effectively implements GOP, whereby the previous bump position of place cells serves as the initial point, which restricts the bump to evolve to a local maximum of the posterior, reflecting the continuity of the animal movement. Hence, no abrupt change in decoding results occurs, which effectively eliminate large non-local errors. Fig. 4b show that as the amplitude of noises in grid cells' responses increases, the error of MAP based on PSC increases dramatically (due to non-local errors), while the error of coupled CANNs remains largely flat.

## 6 CONCLUSIONS AND DISCUSSIONS

In this study, we have developed a computational model to elucidate how place and grid cells complement with each other to improve spatial representation in the brain. We first formulate the interaction between place and grid cells as a probabilistic inference model for integrating information between environmental and motion cues. We then model the interaction between HPC and MEC using reciprocally coupled CANNs, with the coupling strength conveying the correlation prior between sensory cues. We theoretically derive that the dynamics of coupled CANNs effectively implements gradient-based optimization of the posterior (GOP). Using simulations, we further demonstrate that our model indeed achieves Bayesian optimal integration of two sensory cues, which improves the spatial coding accuracy compared to no cue integration. We also show that when only the motion cue is presented, our network model avoids the non-local error problem faced by phase coding of grid cells.

### 6.1 RELATED WORKS

The coding properties of place cells and grid cells have been studied intensively in the literature for long time. For instances, a large volume of theoretical works studied the encoding accuracy and decoding models of place cells (see e.g. (Wilson & McNaughton, 1993; Seung & Sompolinsky, 1993; Brunel & Nadal, 1998; Abbott & Dayan, 1999) and references therein), and researches on grid cells involve calculating the coding efficiency and addressing the non-local error problem (Fiete et al., 2008b; Sreenivasan & Fiete, 2011; Mathis et al., 2012). Notably, Sreenivasan et al. proposed a "constrained range" approach to resolve the non-local error of PSC, which restricts the coding range of grid cells to reduce the sensitivity of phase coding to noises (Sreenivasan

& Fiete, 2011), and they built a network model with connected place cells and grid cells to implement this idea. Their method differs from our model significantly. Specifically, by restricting the spatial range to be decoded, the "constrained range" approach avoids large non-local errors, but it also sacrifices the advantage of high coding efficiency of PSC. On the other hand, in our model, the non-local error of phase coding is naturally avoided, since the coupled CANNs effectively implement a gradient-based searching method (GOP), which imposes that the decoding result of the place cell network is history-dependent and only changes smoothly.

There are also a large volume of theoretical works in the literature studying the interactions between place and grid cells. These works include: 1) exploring how grid cell inputs drive place cells to form localized place fields and how the place field remapping occurs (Bush et al., 2014; Agmon & Burak, 2020; Whittington et al., 2020); 2) exploring how place cell inputs shape the hexagonal activity pattern of grid cells (Blair et al., 2008; Fernandez-Leon et al., 2022; Evans & Burgess, 2019); 3) examining the role of the HPC-MEC loop in multisensory processing. For examples, Li et al. proposed that the reciprocal connections between HPC and MEC transmit the prior association between visual and motion cues, enabling the differentiation of multiple environments (Li et al., 2020); Laptev et al. suggested that attractor dynamics in HPC and MEC enable decentralized information integration (Laptev & Burgess, 2019); Agmon et al. modeled the joint network of grid cells and place cells as coupled CANNs (Agmon & Burak, 2020). Compared to these works, our study has the following novel contributions: 1) we formulate the information integration between HPC and MEC as a Bayesian inference model, and propose GOP as a feasible decoding method; 2) we build a model of coupled CANNs to elucidate the information integration between HPC and MEC, where the reciprocal connections between CANNs convey the prior correlation between sensory cues; 3) by both theoretical analysis and simulations, we demonstrate that the coupled CANNs effectively implement GOP, a Bayesian optimal way of information integration, and that the coupled CANNs eliminate non-local errors of phase coding of grid cells.

## 6.2 LIMITATION AND FUTURE WORK

In the present study, for the purpose of theoretically elucidating the neural mechanism clearly, we have considered the representation of 1D space, while a large amount of experimental data was on the 2D space. Also, the focuses of this work are on the information integration between place and gird cells and eliminating of non-local decoding errors. Experimental data has shown that place and grid cells can also complement with each other on other functions. One example is the construction of a global map of the space (Whittington et al., 2020; Evans et al., 2016). Based on the environmental cue such as vision, place cells can sense the space precisely but only locally; while based on the self-motion cue, gird cells sense the space ambiguously but can link far-away locations via path-integration. Thus, in order to construct a global map, it is necessary to combine both environmental and motion cues. In future work, we will extend the current model to the 2D space and investigate how place and grid cells complement with each other to construct a global spatial map.

Recent experimental studies have continuously revealed new functions of the HPC-MEC loop. In addition to processing different sensory cues, it was found that the spatial map formed by place cells is more oriented to the special layout of the environment; while the spatial map formed by grid cells reflects more the abstract metric information of the environment (Whittington et al., 2020; Evans et al., 2016). This is manifested by that place cells tend to remap their place fields dramatically when the environment changes, while grid cells only realign their place fields slightly (Hafting et al., 2005). Furthermore, it was suggested that in addition to represent the space, the HPC-MEC loop can cooperate to represent some general relationships between objects. It is our future work to extend the current model to explore these issues.

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

## A  APPENDIX A: CODING PROPERTIES OF LSC AND PSC

We compare the different coding properties of place cells and grid cells.

Local spatial coding by place cells, also known as "classical population coding" (Sreenivasan & Fiete, 2011), is widely found in various brain regions (Schreiner et al., 2000; Georgopoulos et al., 1989; Taube et al., 1990; Vollan et al., 2024). In this coding strategy, each neuron has a local receptive field corresponding to its preferred feature values. The receptive field typically has a bell-shaped curve, with neuronal activity peaking when the stimulus is at the center of the receptive field and gradually decreasing as the stimulus moves away from the center. The receptive fields of different neurons cover different parts of the feature space, together spanning the entire encoded variable space. Place cells perceive the animal location primarily through environmental cues, e.g., the visual and/or olfactory cues. Denote the true location of the animal is $z_0$. The response of a place cell with preferred location at $x$ is written as,

$$r_p(x; z_0) = f_p(x; z_0) + \sigma_p \epsilon_p(x) = A_p \exp\left[-\frac{(x-z_0)^2}{4a_p^2}\right] + \sigma_p \epsilon_p(x), \qquad (21)$$

where $f_p(x; z_0)$ is the tuning curve, $A_p$ and $a_p$ the peak and the width of the tuning function, respectively. $\epsilon_p(x)$ denotes Gaussian noise of zero mean and unit variance, and $\sigma_p$ the noise strength. Ideally, the receptive fields of neurons in population coding have identical shapes and are uniformly distributed across the feature space. In this case, each feature value corresponds to a bell-shaped "bump" in neural activity (Figure S1b).

Unlike place cells, grid cells respond to multiple periodic spatial locations. This phenomenon can be equivalently described as grid cells encoding spatial phases. In the one-dimensional space, for grid cells with grid spacing $\lambda$, the mapping between spatial phase and position can be written as (Fiete et al., 2008b),

$$\phi = \psi(z, \lambda) = 2\pi \frac{mod(z, \lambda)}{\lambda}, \tag{22}$$

where $z$ denotes the spatial position, and $\phi$ represents the spatial phase, ranging from 0 to $2\pi$. $\psi(z)$ is the mapping function between position and phase, where $mod$ is the modulus function. Grid cells perceive the animal location primarily through the self-motion cue. Let us denote $\phi^i(z_0)$ the phase of the $i$th module corresponding to the animal location $z_0$. The response of a grid cell in the $i$th module with preferred phase at $\theta^i$ is written as,

$$r_g(\theta^i; \phi^i) = f_g(\theta^i; \phi^i) + \sigma_{gi}\epsilon_g(\theta^i) = A_g \exp\left\{-\frac{||\theta^i - \phi^i||^2}{4a_{gi}^2}\right\} + \sigma_{gi}\epsilon_g(\theta^i), \tag{23}$$

where $f_g(\theta^i; \phi^i)$ is the tuning curve, $A_g$ and $a_{gi}$ the peak and width of the tuning curve, respectively. $\epsilon_g(\theta^i)$ represents Gaussian noise of zero mean and unit variance, and $\sigma_{gi}$ the noise strength. $||\theta^i - \phi^i|| = \min\left(|\theta^i - \phi^i|, 2\pi - |\theta^i - \phi^i|\right)$.

A grid cell module encodes spatial phase at a specific scale using "classical population coding" (Figure S1c). Grid cells from different modules encode spatial phases at different scales (Hafting et al., 2005), collectively forming a phase vector (Figure S1c). This method, known as periodic spatial coding, uses the phase vector encoded by the population activity to represent spatial position (Fiete et al., 2008b).

## A.1 EFFICIENCY AND ROBUSTNESS OF LSC

The robustness of LSC to noise can be characterized by the relationship between noise intensity and decoding error. Previous studies have analytically derived the optimal decoding error for LSC by calculating the Fisher information (Seung & Sompolinsky, 1993; Abbott & Dayan, 1999; Latham et al., 2003; Brunel & Nadal, 1998). These results show that the lower bound of decoding error in LSC is linearly proportional to the variance of the noise and inversely proportional to the total number of place cells. Specifically, under the assumption of independent noises and considering $x \in (-\infty, \infty)$, the Fisher information is calculated to be,

$$J(z) = \rho_p \frac{\sqrt{\pi}A_p^2}{2a_p\sigma_p^2}, \tag{24}$$

where $\rho_p$ represents the density of place cells. We see that the Fisher information increases with the neuron density, indicating that increasing the number of neurons improves the robustness of LSC. This robustness arises from two aspects: first, the bell-shaped bump is formed by the collective activity of many neurons, making it difficult for noise at the single-cell level to disrupt the overall shape of the bump. Second, the similarity between bumps corresponding to different features decreases as the distance between the features increases, making it difficult for single-cell noise to confuse bumps that represent distant feature values (Figure S1b). As a result, noise only causes errors near the actual feature value, rather than introducing random errors.

The trade-off for increased robustness is a decrease in coding efficiency, as more neurons are required to encode the same variable. Inspired by Shannon's information theory (Mackay, 2003), Ila et al. defined the coding efficiency $R$ as the total amount of information encoded (in bits) divided by the number of neurons used for encoding (Sreenivasan & Fiete, 2011). For one-dimensional spatial coding, the total encoded information is proportional to the logarithm of the encoded spatial range $L$, so the coding efficiency $R = \frac{\ln L}{N}$.

Using this definition, we can roughly estimate the efficiency of LSC in the one-dimensional space. Assuming that place cells' place fields uniformly cover a one-dimensional space, let $\Delta x$ be the interval covered by a single place cell, and the total range covered by $N$ neurons is $L = N\Delta x$. Therefore, the coding efficiency is given by:

$$R_p = \frac{\ln(N\Delta x)}{N}. \tag{25}$$

As we can see, as the number of neurons $N$ increases, the efficiency $R_p$ of LSC decreases and asymptotically approaches zero. Compared to PSC (see below), LSC is inefficient in term of utilizing the coding resource of neurons.

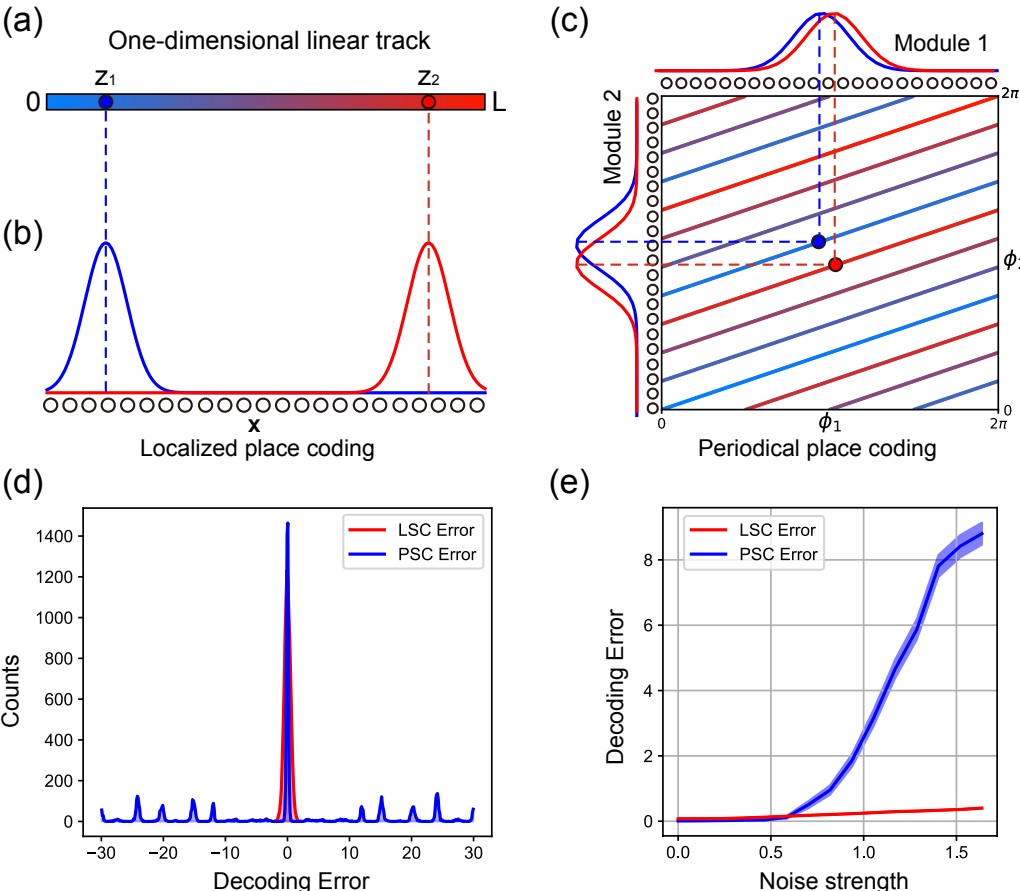

Figure S1: Robustness of local spatial coding and periodic spatial coding. **(a)**. A one-dimensional linear track, where color represents continuously varying positions. The blue and red dots correspond to two distant positions, $z_1$ and $z_2$, respectively. **(b)**. Localized space coding (LSC). Place cells are arranged in a one-dimensional feature space according to their preferred place fields, with neural activity forming a Gaussian bump centered on the encoded position. The blue and red lines represent the bumps encoding the two distinct positions, $z_1$ and $z_2$. **(c)**. Phase space coding (PSC). Spatial positions are encoded by two grid cell modules, forming a two-dimensional phase space represented by the phase combinations $(\phi_1, \phi_2)$. The parallel lines are the projections of the positions in (a) onto the phase space. The blue and red dots correspond to $z_1$ and $z_2$ in (a), with their respective bump activities shown as blue and red lines along the axes. **(d)**. Distribution of decoding errors. The blue line represents PSC, while the red line represents LSC. **(e)**. Relationship between decoding error and noise intensity. The blue line represents PSC, while the red line represents LSC.

## A.2 Efficiency and Robustness of PSC

A single phase can only represents a position within the range of the period $\lambda$, beyond this range, the periodic nature of grid cell activity leads to ambiguity in position representation. To enlarge the representation range, the combination of phases from multiple grid cell modules can be used. If the spatial periods of all grid cell modules are pairwise coprime, the total coding range $\Lambda$ is the product of all periods, i.e., $\Lambda = \prod_i^M \lambda^i = \bar{\lambda}^M$ (Sreenivasan & Fiete, 2011), where $M$ is the number of modules and $\lambda^i$ represents the spatial period of the $i$-th module. Assuming each module contains $N_0$ grid cells, the coding efficiency of PSC is calculated to be:

$$R_g = \frac{\ln \prod_{i=1}^M \lambda^i}{MN_0} = \frac{M \ln \bar{\lambda}}{MN_0} = \frac{\ln \bar{\lambda}}{N_0}. \tag{26}$$

The efficiency of PSC is independent of the number of grid cell modules $M$, which differs from LSC (see Eq. 25). Thus, PSC is more efficient than LSC in term of utilizing the coding resource of neurons.

**Non-local errors in PSC.** The vast coding range of grid cells, which grows exponentially with the number of modules, comes with the expense of high sensitivity to noise. For instance, consider PSC with two grid cell modules. The phase combination $(\phi_1, \phi_2)$ spans a two-dimensional toroidal space with periodic boundaries (Figure S1c). A straight line in position space (with position coordinate $z$) is mapped to parallel line segments in the phase space through the mapping function Eq. (22). The vertical separation between adjacent parallel lines is denoted by $d_{\min}$. It is important to note that, due to the periodicity of the phase space, two points that are far apart in the position space (the blue and red dots in Figure S1c) can be close in the phase space. Consequently, a small perturbation in the phase space can result in a large shift in the decoded positions (Figure S1d) (Sreenivasan & Fiete, 2011).

Specifically, non-local errors occur when noises cause the phase combination encoded by grid cells to deviate by more than $d_{\min}/2$ from the true phase combination in the vertical direction. Since the encoding of phase within a single grid module follows "classical population coding," the error is proportional to the intensity of noises. Therefore, PSC does not produce non-local errors when the noise level is low. However, as the noise level exceeds a certain threshold, non-local errors emerge, leading to a sharp increase in the error variance (as shown in the numerical simulations in Figure S1e). This indicates that PSC is vulnerable to noise.

## B Appendix B: Decoding methods

### B.1 Gradient-based optimization of the posterior (GOP)

In the main text, we have formulated the information integration between place and grid cells as a probabilistic inference problem. The posterior of $z$ and $\phi$ given $\mathbf{r}_g$ and $\mathbf{r}_p$ is expressed as,

$$p(z, \phi | \mathbf{r}_g, \mathbf{r}_p) \propto p(\mathbf{r}_g | \phi) p(\mathbf{r}_p | z) p(\phi, z)$$

$$= \prod_{i=1}^M \left\{ p(\phi^i, z) \prod_{j=1}^{N_0} p\left[r_g(\theta_j^i) | \phi^i\right] \right\} \prod_{j=1}^N p\left[r_p(x_j)|z\right]. \tag{27}$$

Maximizing the posterior is equivalent to maximizing the log posterior, and the latter is given by,

$$\ln p(z, \phi | \mathbf{r}_g, \mathbf{r}_p) \propto \ln p(\mathbf{r}_g | \phi) + \ln p(\mathbf{r}_p | z) + \ln p(\phi, z)$$

$$= -\frac{\rho_g}{\sigma_{gi}^2} \sum_i \int_0^{2\pi} \left[r_g(\theta^i) - f_g(\theta^i; \phi^i)\right]^2 d\theta^i$$

$$- \frac{\rho_p}{\sigma_p^2} \int_{-\infty}^\infty [r_p(x) - f_p(x; z)]^2 dx - \sum_i \frac{1}{\sigma_{\phi^i}^2} ||\phi^i - \psi^i(z)||^2 + C. \tag{28}$$

For the clarity of description, we approximate the discrete neuron distribution as a continuous one, i.e., $\sum_j [r(s_j) - f(s_j)]^2 = \rho_s \int [r(s) - f(s)]^2 ds$. $\rho_g = N_0/2\pi$ represents the density of grid cells and $\rho_p = N_p/L$ the density of place cells. $C = M \ln \sqrt{2\pi} \sigma_{gi} + M \ln \sqrt{2\pi} \sigma_{\phi^i} + N_p \ln \sqrt{2\pi} \sigma_p$ is a constant.

We calculate the gradients of each probability function with respect to $z$ and $\phi^i$. For $p(\phi|z)$, the gradient of $\phi^i$ is calculated to be,

$$\frac{\partial}{\partial \phi^i} \ln p(\phi, z) = \frac{\partial}{\partial \phi^i} \left[ -\frac{||\phi^i - \psi^i(z)||^2}{2\sigma_{\phi^i}^2} \right],$$

$$= \frac{1}{\sigma_{\phi^i}^2} ||\psi^i(z) - \phi^i||, \tag{29}$$

and the gradient of $z$ is,

$$
\begin{aligned}
\frac{\partial}{\partial z} \ln p(\boldsymbol{\phi}, z) &= \frac{\partial}{\partial z} \sum_i \left[ -\frac{||\phi^i - \psi^i(z)||^2}{2\sigma_{\phi^i}^2} \right], \\
&= \sum_i \frac{1}{\sigma_{\phi^i}^2} ||\phi^i - \psi^i(z)|| \frac{\partial \psi^i(z)}{\partial z}, \\
&= \sum_i \frac{2\pi}{\lambda_i \sigma_{\phi^i}^2} ||\phi^i - \psi^i(z)||.
\end{aligned}
\tag{30}
$$

In the above, we have used the von-Mises distribution approximation of $p(\boldsymbol{\phi}, z)$ (see more details below).

For $p(\mathbf{r}_g|\boldsymbol{\phi})$, the gradient of $\phi^i$ is calculated to be,

$$
\begin{aligned}
\frac{\partial}{\partial \phi^i} \ln p(\mathbf{r}_g|\boldsymbol{\phi}) &= \frac{\partial}{\partial \phi^i} \int \rho_g \left[ -\frac{(r_g(\theta^i)) - f_g(\theta^i; \phi^i))^2}{2\sigma_{gi}(\theta^i)} \right] d\theta^i, \\
&= \frac{\rho_g}{\sigma_{gi}^2} \int r_g(\theta^i) \frac{\partial f_g(\theta^i; \phi^i)}{\partial \phi^i} d\theta^i,
\end{aligned}
\tag{31}
$$

The gradient of $z$ is calculated to be,

$$
\frac{\partial}{\partial z} \ln p(\mathbf{r}_g|\boldsymbol{\phi}) = 0.
\tag{32}
$$

For $p(\mathbf{r}_p|z)$, the gradient of $\phi^i$ is calculated to be,

$$
\frac{\partial}{\partial \phi^i} \ln p(\mathbf{r}_p|z) = 0,
\tag{33}
$$

and the gradient of $z$ is,

$$
\begin{aligned}
\frac{\partial}{\partial z} \ln p(\mathbf{r}_p|z) &= \frac{\partial}{\partial z} \rho_p \int \left[ -\frac{(r_p(x) - f_p(x; z))^2}{2\sigma_p^2} \right] dx, \\
&= \frac{\rho_p}{\sigma_p^2} \int r_p(x) \frac{\partial f_p(x; z)}{\partial z} dx.
\end{aligned}
\tag{34}
$$

Combining the above results, the gradients of the log posterior are obtained, which are,

$$
\begin{aligned}
\frac{\partial}{\partial z} \ln p(\boldsymbol{\phi}, z|\mathbf{r}_g, \mathbf{r}_p) &= \frac{\partial}{\partial z} \left[ \ln p(\mathbf{r}_g|\boldsymbol{\phi}) + \ln p(\boldsymbol{\phi}, z) + \ln p(\mathbf{r}_p|z) \right], \\
&= \sum_i \frac{2\pi}{\lambda_i \sigma_{\phi^i}^2} ||\phi^i - \psi^i(z)|| + \frac{\rho_p}{\sigma_p^2} \int r_p(x) \frac{\partial f_p(x; z)}{\partial z} dx,
\end{aligned}
\tag{35}
$$

$$
\begin{aligned}
\frac{\partial}{\partial \phi^i} \ln p(\boldsymbol{\phi}, z|\mathbf{r}_g, \mathbf{r}_p) &= \frac{\partial}{\partial \phi^i} \left[ \ln p(\mathbf{r}_g|\boldsymbol{\phi}) + \ln p(\boldsymbol{\phi}, z) + \ln p(\mathbf{r}_p|z) \right], \\
&= \frac{1}{\sigma_{\phi^i}^2} ||\psi^i(z) - \phi^i|| + \frac{\rho_g}{\sigma_{gi}^2} \int r_g(\theta^i) \frac{\partial f_g(\theta^i; \phi^i)}{\partial \phi^i} d\theta^i.
\end{aligned}
\tag{36}
$$

The algorithm of gradient-based optimization of the posterior (GOP) is summarized in Table 1.

THE VON-MISES DISTRIBUTION APPROXIMATION

In the main text, for convenience, we consider that $p(\boldsymbol{\phi}, z)$ satisfy Gaussian distributions. In reality, von-Mises distributions are more suitable to describe periodic variables. When computing gradients, we use von-Mises distributions to approximate Gaussian distributions, as the latter are not differentiable at the boundary.

Using the von-Mises distribution, the probability $p(\boldsymbol{\phi}, z)$ is written as,

$$
p(\boldsymbol{\phi}, z) = \prod_{i=1}^{M} \frac{1}{2\pi I_0(\kappa)} \exp \left[ \kappa \cos(\phi^i - \psi^i(z)) \right].
\tag{37}
$$

Table 1: The pseudo-code of GOP

| Step | Procedure |
|------|-----------|
| 1 | **Initialization:** Set initial values based on the historical information $z^{(0)} = \hat{z}$, where $\hat{z}$ represents the decoded location at the previous time step. $\phi^{(0)} = \psi(z^{(0)})$ |
| 2 | **Gradient Ascent Update:** Iteratively update parameters using the gradients of the log posterior. Update $z$: $z^{(k+1)} = z^{(k)} + \eta_z \nabla_z \ln p(z^{(k)}, \phi^{(k)} \vert \mathbf{r}_g, \mathbf{r}_p)$ Update $\phi$: $\phi^{(k+1)} = \phi^{(k)} + \eta_\phi \nabla_\phi \ln p(z^{(k)}, \phi^{(k)} \vert \mathbf{r}_g, \mathbf{r}_p)$ where $\nabla_z \ln p$ and $\nabla_\phi \ln p$ are the gradients of the log posterior with respect to $z$ and $\phi$, and $\eta_z$ and $\eta_\phi$ are the learning rates. |
| 3 | **Termination:** Stop iteration when a the local maximum is reached: $\nabla_z \ln p(z^{(k)}, \phi^{(k)} \vert \mathbf{r}_g, \mathbf{r}_p) \approx 0$ and $\nabla_\phi \ln p(z^{(k)}, \phi^{(k)} \vert \mathbf{r}_g, \mathbf{r}_p) \approx 0$. **Output:** $z, \phi$ |

The modified Bessel function of the first kind and zero order $I_o(\kappa)$ serves as the normalizing constant, satisfying $\int_{-\pi}^{\pi} \exp\left[\kappa \cos(x - \mu)\right] = 2\pi I_o(\kappa)$. Parameters $\kappa$ and $\psi(z)$ are analogous to the variance and mean in the Gaussian distribution, respectively.

The logarithm of $p(\phi, z)$ is expressed as,

$$\ln p(\phi, z) = \sum_{i=1}^{M} \kappa \cos(\phi^i - \psi^i(z)) - M \ln(2\pi I_0(\kappa)), \tag{38}$$

where $\psi^i(z) = \mod(z/\lambda^i) \times 2\pi = (z/\lambda^i) \times 2\pi - 2n^i\pi$, with $n^i = z//\lambda^i$ and the symbol $//$ represents the mathematical operation of divisibility. Substituting $\psi^i(z)$ into the above equation, we get,

$$\ln p(\phi, z) = \sum_{i=1}^{M} \kappa \cos(\phi^i - \frac{z}{\lambda^i} 2\pi - 2n^i\pi) + C = \sum_{i=1}^{M} \kappa \cos(\phi^i - \frac{z}{\lambda^i} 2\pi) + C. \tag{39}$$

Thus, the gradient of $z$ is calculated to be,

$$
\begin{aligned}
\frac{\partial}{\partial z} \ln p(\phi, z) &= \frac{\partial}{\partial z} \sum_{i=1}^{M} \kappa \cos(\phi^i - \frac{z}{\lambda^i} 2\pi), \\
&= \sum_i \frac{2\pi\kappa}{\lambda_i} \sin(\phi^i - \frac{z}{\lambda^i} 2\pi), \\
&= \sum_i \frac{2\pi\kappa}{\lambda_i} \sin\left[\phi^i - \left(\frac{z}{\lambda^i} 2\pi + 2n^i\pi\right)\right], \\
&= \sum_i \frac{2\pi\kappa}{\lambda_i} \sin(\phi^i - \psi^i(z)).
\end{aligned}
\tag{40}
$$

Given that the disparity between $\phi^i$ and $\psi^i(z)$ is sufficiently small, we can use the approximation $\sin(x) \approx x$ with $x$ sufficiently small. We have,

$$\frac{\partial}{\partial z} \ln p(\phi, z) = \sum_i \frac{2\pi\kappa}{\lambda_i} \|\phi^i - \psi^i(z)\|. \tag{41}$$

Similarly, the gradient of $\phi^i$ is calculated to be

$$\frac{\partial}{\partial \phi^i} \ln p(\phi, z) = \kappa \|\psi^i(z) - \phi^i\|. \tag{42}$$

These give the results in Eqs. (29,30).

### B.2 DECODING POSITION FROM GRID CELLS

Assuming independent Gaussian noises, the likelihood function of PSC consists of two parts, which are:

$$p(\phi, z) = \prod_{i=1}^{M} p(\phi^i, z) = \frac{1}{\sqrt{2\pi}\sigma_{\phi^i}} \exp\left[\sum_{i=1}^{M} -\frac{||\phi^i - \psi^i(z)||^2}{2\sigma_{\phi^i}^2}\right], \tag{43}$$

$$p(\mathbf{r}_g|\phi) = \prod_{i=1}^{M}\prod_{j=1}^{N_g} p\left[r_g(\theta_j^i)|\phi^i\right] = \frac{1}{\sqrt{2\pi}\sigma_{gi}} \exp\left\{\sum_{i=1}^{M}\sum_{j=1}^{N_0} -\frac{\left[r_g(\theta_j^i) - f_g(\theta_j^i; \phi^i)\right]^2}{2\sigma_{gi}^2}\right\}, \tag{44}$$

where $\theta_j^i$ represents the preferred phase of the $j$th neuron in the $i$th module.

We can us maximum a posterior (MAP) to decode the position, which is written as,

$$\hat{z}, \hat{\phi} = \arg\max_{z,\phi} p(z, \phi|\mathbf{r}_g) = \arg\max_{z,\phi}\left[p(\mathbf{r}_g|\phi)p(\phi, z)\right]. \tag{45}$$

## C APPENDIX C: THEORETICAL ANALYSIS OF THE MODEL DYNAMICS

### C.1 THE MODEL STRUCTURE

We first review the dynamics of our model presented in the main text.

The dynamics of place cells is given by,

$$\tau_p \frac{dU_p(x,t)}{dt} = -U_p(x,t) + \rho_p \int_{-L/2}^{L/2} W_p(x,x')R_p(x',t)dx'$$

$$+ \sum_{i=1}^{M} \rho_g \int_{-\pi}^{\pi} W_{g,p}(x,\theta^i)R_g(\theta^i,t)d\theta^i + I_p(x). \tag{46}$$

The dynamics of grid cells is given by,

$$\tau_g \frac{dU_g(\theta^i,t)}{dt} = -U_g(\theta^i,t) + \rho_g \int_{-\pi}^{\pi} W_g(\theta^i,\theta^{i'})R_g(\theta^{i'},t)d\theta^{i'}$$

$$+ \rho_p \int_{-L/2}^{L/2} W_{g,p}(x,\theta^i)R_p(x,t)dx + I_g^i(\theta^i). \tag{47}$$

The firing rate of neurons is given by,

$$R_s(s,t) = \frac{U_s(s,t)^2}{1 + k_s\rho_s \int U_s(s,t)^2 ds}, \quad s = x, \theta. \tag{48}$$

The recurrent connections between neurons in the P-CANN or a G-CANN are given by

$$W_s(s,s') = \frac{J_s}{\sqrt{2\pi}a_s} \exp\left[-\frac{||s-s'||^2}{2a_s^2}\right], \quad s = x, \theta. \tag{49}$$

The reciprocal connections between place and grid cells are given by,

$$W_{g,p}(x,\theta) = \frac{J_{g,p}}{\sqrt{2\pi}a_{gi,p}} \exp\left[-\frac{||\theta - \psi(x)||^2}{2a_{gi,p}^2}\right], \tag{50}$$

where $\psi(x) = \mod(x/\lambda, 1) \times 2\pi$.

### C.2 THE SOLUTION OF BUMP STATES

When the inhibition strength $k$ lies within a certain region (Fung et al., 2010), a CANN can hold a continuous family of bump-shape stationary states. For place cells, the bump state is expressed as,

$$\bar{U}_p(x) = A_p \exp\left[-\frac{(x-z)^2}{4a_p^2}\right], \quad \bar{R}_p(x) = \hat{R}_p \exp\left[-\frac{(x-z)^2}{2a_p^2}\right]. \tag{51}$$

For grid cells, the bump state is expressed as,

$$\bar{U}_g(\theta^i) = A_g \exp\left[-\frac{||\theta^i - \phi^i||^2}{4a_p^2}\right], \quad \bar{R}_g(\theta^i) = \hat{R}_g \exp\left[-\frac{||\theta^i - \phi^i||^2}{2a_{gi}^2}\right]. \tag{52}$$

In the above $A_p$ and $A_g$ represent the bump heights in P-CANN and G-CANN, respectively, and $\hat{R}_p$ and $\hat{R}_g$ the maximum firing rates of place cells and grid cells, respectively. We check the condition when the coupled networks hold these bump states as their stationary states. For clearance, hereafter, we denote the Gaussian function as $\mathcal{N}(s|s', a) = \exp\left[-||s - s'||^2/4a^2\right]$.

Substituting Eqs. (51 ,52) into the grid cell dynamics, we get,

$$\tau_g\left[A_g \frac{||\theta^i - \phi^i||}{2a_{gi}^2}\frac{d\phi^i}{dt} + \tau_g \frac{dA_g}{dt}\right]\mathcal{N}(\theta|\phi^i, a_{gi}) = (-A_g + \frac{\rho_g J_g}{\sqrt{2}}\hat{R}_g)\mathcal{N}(\theta|\phi^i, a_{gi})$$
$$+ \rho_p \int \frac{J_{g,p}\hat{R}_p}{\sqrt{2\pi}a_{gi,p}}\mathcal{N}(\theta|\psi(x), \frac{a_{gi,p}}{\sqrt{2}})\mathcal{N}(x|z, \frac{a_p}{\sqrt{2}})]dx + I_g^i(\theta^i, t), \tag{53}$$

where $\mathcal{N}(\theta|\phi^i, a_{gi}) = \exp\left[-||\theta^i - \phi^i||^2/4a_{gi}^2\right]$. By setting $I_g^i = 0$ and $dA_g/dt = 0$, the dynamics of grid cells can be simplified as,

$$(A_g - \frac{\rho_g J_g}{\sqrt{2}}\hat{R}_g)\mathcal{N}(\theta|\phi^i, a_{gi}) = \rho_p \int \frac{J_{g,p}\hat{R}_p}{\sqrt{2\pi}a_{gi,p}}\mathcal{N}(\theta|\psi(x), \frac{a_{gi,p}}{\sqrt{2}})\mathcal{N}(x|z, \frac{a_p}{\sqrt{2}})dx. \tag{54}$$

To calculate the integral in the right-hand side of Eq. (54), we need to map the bump from the phase representation to the location representation. Because of the periodic nature, a phase corresponds to many locations. Denote $\psi^{-1}(\theta^i; z) = (\lambda^i/2\pi)\theta^i + n^i(z)$ to be the location corresponding to phase $\theta$ closest to $z$. We have,

$$\mathcal{N}(\theta^i|\psi^i(z), a_{gi}) = \exp\left[-\frac{||\theta^i - \psi^i(z)||^2}{2a_{gi}^2}\right],$$
$$= \exp\left[-\frac{||\theta^i - 2\pi(z/\lambda^i - n^i)||^2}{2a_{gi}^2}\right],$$
$$= \exp\left[-\frac{((\lambda^i/2\pi)\theta^i + n^i(z) - z)^2}{2a_p^2}\right], \tag{55}$$
$$= \exp\left[-\frac{(\psi^{-1}(\theta^i; z) - z)^2}{2a_p^2}\right],$$
$$= \mathcal{N}(z|\psi^{-1}(\theta^i; z), a_p).$$

Note that $|\psi^{-1}(\theta^i; z) - z|$ must be smaller than $\lambda^i/2$, implying that $||\theta^i - 2\pi(z/\lambda^i - n^i(z))|| = |\theta^i - 2\pi(z/\lambda^i - n^i(z))| < \pi$, so that the operation of circular distance can be removed from the above equation.

Using this property, we can further simplify Eq. (54) as,

$$(A_g - \frac{\rho_g J_g}{\sqrt{2}}\hat{R}_g)\mathcal{N}(\theta|\phi^i, a_{gi}) = \rho_p \frac{J_{g,p}\hat{R}_p}{\sqrt{2\pi}a_{gi,p}}\int \mathcal{N}(\theta|\psi(x), \frac{a_{gi,p}}{\sqrt{2}})\mathcal{N}(x|z, \frac{a_p}{\sqrt{2}})dx,$$
$$= \rho_p \frac{J_{g,p}\hat{R}_p}{\sqrt{2\pi}a_{gi,p}}\int \mathcal{N}(z|\psi^{-1}(\theta; x), \frac{\lambda}{2\pi}\frac{a_{gi,p}}{\sqrt{2}})\mathcal{N}(x|z, \frac{a_p}{\sqrt{2}})dx. \tag{56}$$

We see that, the condition for the above equation to be satisfied is,

$$a_{gi,p} = a_{gi} = a_p(2\pi/\lambda), \tag{57}$$

We substitute this condition into the above equation and get,

$$(A_g - \frac{\rho_g J_g}{\sqrt{2}}\hat{R}_g)\mathcal{N}(\theta|\phi^i, a_{gi}) = \frac{\rho_p J_{g,p}\hat{R}_p}{\sqrt{2}}\mathcal{N}(x|\psi^{-1}(\theta; x), a_p),$$
$$= \frac{\rho_p J_{g,p}\hat{R}_p\lambda}{2\sqrt{2\pi}}\mathcal{N}(\theta|\psi(z), a_{gi}). \tag{58}$$

Therefore, in the steady state, the bump center of grid cell modules is $\phi^i = \psi(z)$, with $z$ the bump center of place cells.

Substituting Eqs. (51 ,52) into the place cell dynamics, we get,

$$\tau_p A_p \frac{x-z}{2a_p^2}\mathcal{N}(z, 2a_p^2)\frac{dz}{dt} + \tau_p \frac{dA_p}{dt}\mathcal{N}(z, 2a_p^2) = (-A_p + \frac{\rho_p J_p}{\sqrt{2}}\hat{R}_p)\mathcal{N}(z, 2a_p^2)$$
$$+ \sum_{i=1}^{M} \frac{\rho_g J_{g,p}\hat{R}_g}{\sqrt{2}}\exp\left[-\frac{||\psi(x)-\phi^i||^2}{4a_{gi}^2}\right] + I_p(x,t). \tag{59}$$

The divisive normalization can be simplified as,

$$\hat{R}_s = \frac{A_s^2}{1 + \sqrt{2\pi}a_s\rho_s k_s A_s^2}, \tag{60}$$

where $s$ denotes the cell type.

We calculate the stationary state of the networks when no external input exist ($\mathbf{I}_p = \mathbf{I}_g^i = 0$). By setting $dA_p/dt = dA_g/dt = 0$ and $dz/dt = d\phi^i/dt = 0$, we obtain

$$A_p = \frac{\rho_p J_p}{\sqrt{2}}\hat{R}_p + \frac{1}{\sqrt{2}}\sum_i \rho_g \hat{R}_g J_{g,p}, \tag{61}$$

$$A_g = \frac{\rho_g J_g}{\sqrt{2}}\hat{R}_g + \frac{1}{\sqrt{2}}\rho_p R_p J_{g,p}. \tag{62}$$

Using divisive normalization, we get,

$$A_g = \frac{\rho_g J_g}{\sqrt{2}}\frac{A_g^2}{1 + \sqrt{2\pi}a_{gi}\rho_g k_g A_g^2} + \frac{\rho_p J_{g,p}}{\sqrt{2}}\frac{A_p^2}{1 + \sqrt{2\pi}a_p\rho_p k_p A_p^2}, \tag{63}$$

$$A_p = \frac{\rho_p J_p}{\sqrt{2}}\frac{A_p^2}{1 + \sqrt{2\pi}a_p\rho_p k_p A_p^2} + \frac{1}{\sqrt{2}}\sum_i \frac{\rho_g J_{g,p}a_p}{a_{gi}}\frac{A_g^2}{1 + \sqrt{2\pi}a_{gi}\rho_g k_g A_g^2}. \tag{64}$$

Previous research has shown that for CANNs, external inputs primarily influence the network dynamics by altering the position of the network's activity bump (Wong et al., 2008; Fung et al., 2010). In our model, this corresponds to the effects on $dz/dt$ and $d\phi^i/dt$. This sensitivity arises because CANNs exhibit neutral stability in the direction of the activity bump's movement, making them highly responsive to external inputs. In contrast, the height of the activity bump is less sensitive to changes in external inputs, reflecting the inherent dynamics of the attractor. When examining changes in bump height, we consider that recurrent inputs within each network are much stronger than reciprocal ones, allowing us to disregard reciprocal inputs and simplify the above expressions,

$$A_g = \frac{\rho_g J_g}{\sqrt{2}}\frac{A_g^2}{1 + \sqrt{2\pi}a_{gi}\rho_g k_g A_g^2}, \tag{65}$$

$$A_p = \frac{\rho_p J_p}{\sqrt{2}}\frac{A_p^2}{1 + \sqrt{2\pi}a_p\rho_p k_p A_p^2}. \tag{66}$$

We have,

$$A_g = \frac{1}{4\sqrt{\pi}a_{gi}\rho_g k_g}\left(\rho_g J_g + \sqrt{\rho_g^2 J_g^2 - 8\sqrt{2\pi}a_{gi}\rho_g k_g}\right), \tag{67}$$

$$A_p = \frac{1}{4\sqrt{\pi}a_p\rho_p k_p}\left(\rho_p J_p + \sqrt{\rho_p^2 J_p^2 - 8\sqrt{2\pi}a_p\rho_p k_p}\right). \tag{68}$$

### C.3 SIMPLIFYING THE MODEL DYNAMICS BY THE PROJECTION METHOD

In a CANN, its stationary states constitute a neutrally stable sub-manifold, implying that the network dynamics can be effectively represented by a small number of dominant motion modes, such as the variations in height and position of the stationary states. Therefore, by projecting the network dynamics onto these dominant motion modes, we can simplify the network dynamics significantly. The first two dominating motion modes of a CANN can be expressed as,

$$\text{height} \quad : \quad u_s^0(s) = \bar{U}_s(s), \tag{69}$$

$$\text{position} \quad : \quad u_s^1(s) = \frac{\partial \bar{U}_s(s)}{\partial s}. \tag{70}$$

### C.3.1 THE SIMPLIFIED DYNAMICS OF PLACE CELLS

For place cells, the first two modes are written as,

$$u_p^0(x) = \exp\left[-\frac{(x-z)^2}{4a_p^2}\right], \tag{71}$$

$$u_p^1(x) = [x-z]\exp\left[-\frac{(x-z)^2}{4a_p^2}\right]. \tag{72}$$

We consider that the field width of place cells is much smaller than the space range, i.e., $a_p \ll L$, and the integral from $-L/2$ to $L/2$ can be approximate as from $-\infty$ to $\infty$.

For the projection on $u_p^0$, the result is,

$$
\begin{aligned}
\text{Left} &= \int \tau_p \frac{A_p}{2a_p^2}\frac{dz}{dt}(x-z)\exp\left[-\frac{(x-z)^2}{4a_p^2}\right]u_p^0(x)dx + \int \tau_p\frac{dA_p}{dt}\exp\left[-\frac{(x-z)^2}{4a_p^2}\right]u_p^0(x)dx, \\
&= \sqrt{2\pi}a_p\tau_p\frac{dA_p}{dt}.
\end{aligned}
\tag{73}
$$

$$
\begin{aligned}
\text{Right} &= \int\left(-A_p+\frac{\rho_p J_p}{\sqrt{2}}\hat{R}_p\right)\exp\left[-\frac{(x-z)^2}{4a_p^2}\right]u_p^0(x)dx \\
&\quad + \int\sum_i\frac{\rho_g\hat{R}_g J_{g,p}}{\sqrt{2}}\exp\left[-\frac{||\phi^i-\psi^i(x)||^2}{4a_{gi}^2}\right]u_p^0(x)dx + \int I_{\mathsf{p}}(x,t)u_p^0(x)dx, \\
&= \sqrt{2\pi}a_p\left(-A_p+\frac{\rho_p J_p}{\sqrt{2}}\hat{R}_p\right)+\sqrt{\pi}a_p\sum_i\rho_g\hat{R}_g J_{g,p}\exp\left[-\frac{||\phi^i-\psi^i(z)||^2}{8a_{gi}^2}\right]+\int I_p u_p^0 dx.
\end{aligned}
\tag{74}
$$

For the projection on $u_p^1$, the result is,

$$
\begin{aligned}
\text{Left} &= \int \tau_p A_p\frac{x-z}{2a_p^2}\frac{dz}{dt}\exp\left[-\frac{(x-z)^2}{4a_p^2}\right]u_p^1(x)dx \\
&\quad + \int \tau_p\frac{dA_p}{dt}\exp\left[-\frac{(x-z)^2}{4a_p^2}\right]u_p^1(x)dx, \\
&= 2\sqrt{\pi}a_p\tau_p A_p\frac{dz}{dt}.
\end{aligned}
\tag{75}
$$

$$
\begin{aligned}
\text{Right} &= \int\left(-A_p+\frac{\rho_p J_p}{\sqrt{2}}\hat{R}_p\right)\exp\left[-\frac{(x-z)^2}{4a_p^2}\right]u_p^1(x)dx \\
&\quad + \int\sum_i\frac{\rho_g\hat{R}_g J_{g,p}}{\sqrt{2}}\exp\left[-\frac{[x-\psi^{-1}(\phi^i)]^2}{4a_p^2}\right]u_p^1(x)dx + \int I_{\mathsf{p}}(x,t)u_p^1(x)dx \\
&= \frac{\sqrt{\pi}a_p}{2}\sum_i\frac{\lambda_i J_{g,p}\rho_g\hat{R}_g}{2\pi}||\phi^i-\psi^i(z)||\exp\left[-\frac{||\psi^i(z)-\phi^i||^2}{8a_{gi}^2}\right]+\int I_p u_p^1 dx
\end{aligned}
\tag{76}
$$

Combining the above equalities, we obtain

$$\frac{dA_p}{dt} = \frac{1}{\tau_p}\left\{-A_p+\frac{\rho_p J_p}{\sqrt{2}}\hat{R}_p+\frac{1}{\sqrt{2}}\sum_i\rho_g\hat{R}_g J_{g,p}\exp\left[-\frac{||\psi^i(z)-\phi^i||^2}{8a_{gi}^2}\right]+\frac{1}{\sqrt{2\pi}a_p}\int I_p u_p^0 dx\right\}, \tag{77}$$

$$\frac{dz}{dt} = \frac{1}{\tau_p A_p}\left\{\frac{1}{4}\sum_i\frac{\lambda_i}{2\pi}J_{g,p}\rho_g\hat{R}_g||\phi^i-\psi^i(z)||\exp\left[-\frac{||\psi^i(z)-\phi^i||^2}{8a_{gi}^2}\right]+\frac{1}{2\sqrt{\pi}a_p}\int I_p u_p^1 dx\right\}. \tag{78}$$

### C.3.2 THE SIMPLIFIED DYNAMICS OF GRID CELLS

For grid cells, the first two dominant motion modes are written as,

$$u_g^0(\theta^i) \quad = \quad \exp\left[-\frac{||\theta^i - \phi^i||^2}{4a_{gi}^2}\right], \tag{79}$$

$$u_g^1(\theta^i) \quad = \quad ||\theta^i - \phi^i|| \exp\left[-\frac{||\theta^i - \phi^i||^2}{4a_{gi}^2}\right]. \tag{80}$$

For the projection on $u_g^0$, the result is,

$$
\begin{aligned}
\text{Left} &= \int \left\{ \tau_g A_g \frac{||\theta^i - \phi^i||}{2a_{gi}^2} \exp\left[-\frac{||\theta^i - \phi^i||^2}{4a_{gi}^2}\right] \frac{d\phi^i}{dt} \right\} u_g^0(\theta^i) d\theta^i \\
&\quad + \int \left\{ \tau_g \frac{dA_g}{dt} \exp\left[-\frac{||\theta^i - \phi^i||^2}{4a_{gi}^2}\right] \right\} u_g^0(\theta^i) d\theta^i, \\
&= \sqrt{2\pi} a_{gi} \tau_g \frac{dA_g}{dt}.
\end{aligned}
\tag{81}
$$

$$
\begin{aligned}
\text{Right} &= \int \left(-A_g + \frac{\rho_g J_g}{\sqrt{2}}\right) \hat{R}_g \exp\left[-\frac{||\theta^i - \phi^i||^2}{4a_{gi}^2}\right] u_g^0(\theta^i) d\theta^i \\
&\quad + \int \frac{\rho_p \hat{R}_p J_{g,p}}{\sqrt{2}} \frac{\lambda_i}{2\pi} \exp\left[-\frac{||\theta^i - \psi^i(z)||^2}{4a_{gi}^2}\right] u_g^0(\theta^i) d\theta^i + \int I_g(\theta^i, t) u_g^0(\theta^i) d\theta^i, \\
&= \sqrt{2\pi} a_{gi} \left(-A_g + \frac{\rho_g J_g}{\sqrt{2}} \hat{R}_g\right) + \sqrt{\pi} a_p \rho_p \hat{R}_p J_{g,p} \exp\left[-\frac{||\phi^i - \psi^i(z)||^2}{8a_{gi}^2}\right] + \int I_g^i u_g^0 dx.
\end{aligned}
\tag{82}
$$

For the projection on $u_g^1$, the result is,

$$
\begin{aligned}
\text{Left} &= \int \left\{ \tau_g A_g \frac{||\theta^i - \phi^i||}{2a_{gi}^2} \exp\left[-\frac{||\theta^i - \phi^i||^2}{4a_{gi}^2}\right] \frac{d\phi^i}{dt} \right\} u_g^1(\theta^i) d\theta^i \\
&\quad + \int \left\{ \tau_g \frac{dA_g}{dt} u_g^0(\theta^i) \right\} u_g^1(\theta^i) d\theta^i, \\
&= 2\sqrt{\pi} a_{gi} \tau_g A_g \frac{d\phi^i}{dt}.
\end{aligned}
\tag{83}
$$

$$
\begin{aligned}
\text{Right} &= \int \left(-A_g + \frac{\rho_g J_g}{\sqrt{2}} \hat{R}_g\right) \exp\left[-\frac{||\theta^i - \phi^i||^2}{4a_{gi}^2}\right] u_g^1(\theta^i) d\theta^i \\
&\quad + \int \frac{a_p}{a_{gi}} \frac{\rho_p \hat{R}_p J_{g,p}}{\sqrt{2}} \exp\left[-\frac{||\theta^i - \psi^i(z)||^2}{4a_{gi}^2}\right] u_g^1(\theta^i) d\theta^i \\
&\quad + \int I_g(\theta^i, t) u_g^1(\theta^i) d\theta^i, \\
&= \frac{\sqrt{\pi} a_p}{2} J_{g,p} \rho_p \hat{R}_p ||\phi^i - \psi^i(z)|| \exp\left[-\frac{||\psi^i(z) - \phi^i||^2}{8a_{gi}^2}\right] + \int I_g^i u_g^1 dx.
\end{aligned}
\tag{84}
$$

Synthesizing the above equalities, we obtain

$$\frac{dA_g}{dt} = \frac{1}{\tau_g} \left\{ -A_g + \frac{\rho_g J_g}{\sqrt{2}} \hat{R}_g + \frac{1}{\sqrt{2}} \rho_p \hat{R}_p J_{g,p} \exp\left[-\frac{||\phi^i(t) - \psi^i(z)||^2}{8a_{gi}^2}\right] + \frac{1}{\sqrt{2\pi} a_{gi}} \int I_g^i u_g^0 dx \right\}. \tag{85}$$

$$\frac{d\phi^i}{dt} = \frac{1}{\tau_g A_g} \left\{ \frac{\lambda}{8\pi} J_{g,p} \rho_p \hat{R}_p ||\psi^i(z) - \phi^i|| \exp\left[-\frac{||\phi^i(t) - \psi^i(z)||^2}{8a_{gi}^2}\right] + \frac{1}{2\sqrt{\pi} a_{gi}} \int I_g^i u_g^1 dx \right\}. \tag{86}$$

Combing the above results and assuming that the network bump centers are close enough, i.e., $||\phi^i(t) - \psi^i(z)||$ are sufficiently small, we get the final form of the simplified network dynamics, which are,

$$\frac{dA_p}{dt} = \frac{1}{\tau_p} \left\{ -A_p + \frac{\rho_p J_p}{\sqrt{2}} \hat{R}_p + \frac{1}{\sqrt{2}} \sum_i \rho_g \hat{R}_g J_{g,p} + \frac{1}{\sqrt{2\pi} a_p} \int I_p u_p^0 dx \right\}, \tag{87}$$

$$\frac{dz}{dt} = \frac{1}{\tau_p A_p} \left\{ \frac{1}{4} \sum_i \frac{\lambda_i}{2\pi} J_{g,p} \rho_g \hat{R}_g ||\phi^i - \psi^i(z)|| + \frac{1}{2\sqrt{\pi} a_p} \int I_p u_p^1 dx \right\} \tag{88}$$

$$\frac{dA_g}{dt} = \frac{1}{\tau_g} \left\{ -A_g + \frac{\rho_g J_g}{\sqrt{2}} \hat{R}_g + \frac{1}{\sqrt{2}} \rho_p \hat{R}_p J_{g,p} + \frac{1}{\sqrt{2\pi} A_g} \int I_g^i u_g^0 dx \right\}, \tag{89}$$

$$\frac{d\phi^i}{dt} = \frac{1}{\tau_g A_g} \left\{ \frac{\lambda}{8\pi} J_{g,p} \rho_p \hat{R}_p ||\psi^i(z) - \phi^i|| + \frac{1}{2\sqrt{\pi} a_{gi}} \int I_g^i u_g^1 dx \right\}. \tag{90}$$

## C.4 The coupled networks implementing GOP

From Eqs.(35,36), the gradients of the logarithm posterior are given by,

$$\frac{\partial}{\partial z} \ln p(\phi, z | \mathbf{r}_g, \mathbf{r}_p) = \sum_i \frac{2\pi}{\lambda_i \sigma_{\phi^i}^2} ||\phi^i - \psi^i(z)|| + \frac{\rho_p}{\sigma_p^2} \int r_p(x) \frac{\partial f_p(x; z)}{\partial z} dx, \tag{91}$$

$$\frac{\partial}{\partial \phi^i} \ln p(\phi, z | \mathbf{r}_g, \mathbf{r}_p) = \frac{1}{\sigma_{\phi^i}^2} ||\psi^i(z) - \phi^i|| + \frac{\rho_g}{\sigma_{gi}^2} \int r_g(\theta^i) \frac{\partial f_g(\theta^i; \phi^i)}{\partial \phi^i} d\theta^i. \tag{92}$$

From Eqs.(88,90), the dynamics of the bump centers of place and grid cells are written as,

$$\frac{dz}{dt} = \frac{1}{\tau_p A_p} \left\{ \frac{1}{4} \sum_i \frac{\lambda_i}{2\pi} J_{g,p} \rho_g \hat{R}_g ||\phi^i - \psi^i(z)|| + \frac{1}{2\sqrt{\pi} a_p} \int I_p u_p^1 dx \right\}, \tag{93}$$

$$\frac{d\phi^i}{dt} = \frac{1}{\tau_g A_g} \left\{ \frac{\lambda}{8\pi} J_{g,p} \rho_p \hat{R}_p ||\psi^i(z) - \phi^i|| + \frac{1}{2\sqrt{\pi} a_{gi}} \int I_g^i u_g^1 d\theta \right\}. \tag{94}$$

We consider the external inputs to the networks are given by, $\mathbf{I}_g^i = \alpha_{gi} \mathbf{r}_g$ and $\mathbf{I}_p = \alpha_p \mathbf{r}_p$, where $\alpha_{gi}$ and $\alpha_p$ represent the strengths of external inputs to the grid cells and place cells, respectively.

Comparing the gradients of the posterior Eqs. (91,92) with the network dynamics Eqs. (93,94), we see that they are equivalent if the below parameter conditions are satisfied, which are,

$$(\frac{\lambda_i}{2\pi})^2 \frac{J_{g,p} \rho_g \hat{R}_g}{4 A_p \tau_p} = \frac{1}{\sigma_{\phi^i}^2}, \quad \frac{a_p \alpha_p}{\sqrt{\pi} A_p^3 \rho_p \tau_p} = \frac{1}{\sigma_p^2},$$

$$\frac{\lambda_i}{2\pi} \frac{J_{g,p} \rho_p \hat{R}_p}{4 A_g \tau_g} = \frac{1}{\sigma_{\phi^i}^2}, \quad \frac{a_{gi} \alpha_{gi}}{\sqrt{\pi} A_g^3 \rho_g \tau_g} = \frac{1}{\sigma_{gi}^2}. \tag{95}$$

From the above equalities, we find the necessary condition for $\sigma_{\phi^i}$ well-defined is $2\pi A_p \tau_p \rho_p \hat{R}_p = \lambda_i A_g^i \tau_g \rho_g^i \hat{R}_g^i$. The relationships between the parameters in the probabilistic model and those in the network dynamics are: $\sigma_p^2 = (\sqrt{\pi} A_p^3 \rho_p \tau_p)/(2a_p \alpha_p)$, $\sigma_{\phi^i}^2 = (8\pi A_g \tau_g)/(\lambda_i J_{g,p} \rho_p \hat{R}_p)$ and $\sigma_{gi}^2 = (\sqrt{\pi} A_g^3 \rho_g \tau_g)/(2a_{gi} \alpha_{gi})$.

For the sake of simulation convenience, we can simplify the above parameter constraint by decomposing them into a more manageable form. Assuming the neuron densities and recurrent strength of grid cells and place cells are equal, i.e. $\rho_p = \rho_g$, $J_g = J_p$, and that the products of interaction width and global inhibition strength within each type of cells are the same, $a_{gi} k_g = a_p k_p$. Therefore, according to the previous derivation of height of stationary synaptic input $A_s$ and maximal firing rate $\hat{R}_s$, we have

$$A_s = \frac{\rho_s J_s}{\sqrt{2}} \frac{A_s^2}{1 + \sqrt{2\pi} a_s \rho_s k_g A_s^2} + \frac{\rho_p J_{g,p}}{\sqrt{2}} \frac{A_s^2}{1 + \sqrt{2\pi} a_s \rho_p k_p A_s^2}, \tag{96}$$

$$\hat{R}_s = \frac{A_s^2}{1 + \sqrt{2\pi} a_s \rho_s k_s A_s^2}, \tag{97}$$

where s denotes the cell type. We see that $A_p = A_g$ and $\hat{R}_p = \hat{R}_g$. Substituting into well-defined condition of $\sigma_{\phi^i}$, $2\pi A_p \tau_p \rho_p \hat{R}_p = \lambda_i A_g^i \tau_g \rho_g^i \hat{R}_g^i$, we can conclude that $a_p \tau_p = a_{gi} \tau_g$. Utilizing the relation between interaction width $a_{gi}/2\pi = a_p/\lambda_i$, we can decompose the constrain as

$$A_p = A_g, \quad J_p = J_g, \quad \frac{k_p}{2\pi} = \frac{k_g}{\lambda_i}, \quad \tau_p = \tau_g \frac{\lambda_i}{2\pi}. \tag{98}$$

Through simulations, we directly compared the evolution trajectory of the bump center when projected with noisy inputs to the trajectory obtained using gradient-based optimization of the posterior (GOP). The results demonstrate that, while there is a small difference in the final converged point between the two decoding methods (which is not statistically significant, see Fig. 3b), the evolution trajectories of the network's bump center and the GOP method are nearly identical (see Fig. S2 for examples).

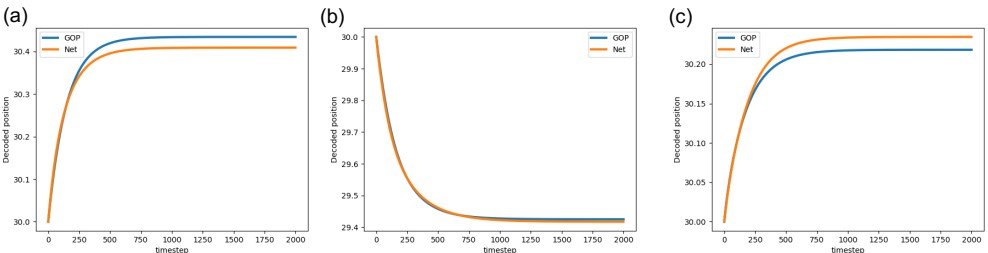

Figure S2: Three examples of decoding trajectories comparing the network's bump center and the GOP method.

# D    APPENDIX D: SIMULATION DETAILS

In this section, we provide the details of the simulation experiments. In simulations, we consider a 1D spatial range of $(-30, 30)$, with the animal located at position $0$. The initial states of the place cell and grid cell CANNs are set to encode the position $z_0 = -0.5$, which is assumed to be the animal's previous location. Most parameters remain consistent across all experiments, with only the input strength of place and grid cells, as well as the variance of noisy inputs, varying between experiments. Below, we present the shared parameters. The parameters for the probabilistic inference model are listed in Table 2, the parameters for place cell dynamics in Table 3, and the parameters for grid cell dynamics in Table 4. Experiment-specific parameters are provided in the subsequent subsections.

| Variable | Value | Variable | Value |
|---|---|---|---|
| $L$ | 60 | $\sigma_{\phi^1}$ | 0.25 |
| $\sigma_{\phi^2}$ | 0.19 | $\sigma_{\phi^3}$ | 0.15 |

Table 2: Shared parameters in the probabilistic inference model.

| Variable | Value | Variable | Value | Variable | Value |
|---|---|---|---|---|---|
| $\rho_p$ | 3.3 | $N_p$ | 200 | $a_p$ | 0.3 |
| $J_p$ | 20 | $k_p$ | 20 | $\tau_p$ | 1.0 |

Table 3: Shared parameters of the place cell dynamics.

## D.1    SIMULATION EXPERIMENT: OPTIMAL INFORMATION INTEGRATION OF SENSORY CUES

The experiment in Sec. 5.1 aims to demonstrate that the network model optimally integrates location information from both environmental and motion cues. We evaluate the network's performance under three distinct conditions:

- **Environmental cue only:** Place cells receive external input from the environmental cue, with input strength $\alpha_p = 0.05$ and grid cell input strength $\alpha_g = 0$.

| Variable | Value | Variable | Value | Variable | Value |
|---|---|---|---|---|---|
| $\rho_g$ | 3.3 | $N_g$ | 20 | $M$ | 3 |
| $\lambda_1$ | 3 | $\lambda_2$ | 4 | $\lambda_3$ | 5 |
| $J_g$ | 20 | $J_{pg}$ | 0.5 | | |
| $a_{\phi^1}$ | 0.63 | $a_{\phi^2}$ | 0.47 | $a_{\phi^3}$ | 0.38 |
| $k_{g,1}$ | 9.55 | $k_{g,2}$ | 12.73 | $k_{g,3}$ | 15.91 |
| $\tau_{g,1}$ | 2.09 | $\tau_{g,2}$ | 1.57 | $\tau_{g,3}$ | 1.26 |

Table 4: Shared parameters of the grid cell dynamics.

- **Motion cue only:** Grid cells receive external input from the motion cue, with $\alpha_p = 0$ and $\alpha_g = 0.05$.
- **Information integration:** Both place and grid cells receive external inputs, with $\alpha_p = 0.05$ from the environmental cue and $\alpha_g = 0.05$ from the motion cue.

External inputs to place and grid cells are generated according to Eqs. (1) and (23), respectively.

For each condition, we conduct 1000 independent trials. In each trial, the animal's location is decoded based on the bump center of place cells, which is computed as:

$$z = \frac{\sum_i R_p(x_i) x_i}{\sum_i R_p(x_i)}, \tag{99}$$

where $R_p(x_i)$ represents the activity of place cells at position $x_i$.

For comparison, we also apply the GOP method (Table 1) to theoretically decode the animal's position.

In all three cases, we calculate the distributions of decoding errors and fit them with Gaussian distributions. In Fig. 3a, the standard deviation of the place cell input is $\sigma_p = 0.25$ and the standard deviation of the grid cell input is $\sigma_g = 0.2$. In Fig. 3b, the standard deviations of the place and grid cell inputs are identical, ranging from 0.05 to 0.2.

### D.2  SIMULATION EXPERIMENTS: ELIMINATING NON-LOCAL ERRORS

The experiment in Sec. 5.2 aims to demonstrate that the network model can effectively eliminate non-local errors associated with phase coding of grid cells. In this setup, only grid cells receive external inputs, meaning that only the motion cue is available (i.e., $\alpha_p = 0$, $\alpha_g = 0.05$).

We compute the network's decoding results and compare them with those obtained using the Maximum A Posteriori (MAP) approach (see Eq.(45)). First, we measure the distributions of decoding errors from both methods at a fixed noise intensity (Fig.4a, $\sigma_g = 0.2$), and then we calculate the variances of the decoding errors across different noise levels (Fig.4b, with $\sigma_g$ ranging from 0.05 to 04). The results are summarized in Fig.4 in the main text.

## E  APPENDIX E: ROBUSTNESS TEST

### E.1  IMPACT OF CORRELATED NOISE

We conduct the same experiment as shown in Fig.3 and Fig.4 in the main text but with correlated multi-variate Gaussian noise, whose covariance matrix is,

$$\langle (r_i - f_i)(r_j - f_j) \rangle = \sigma^2 \big[ \delta_{ij} + c(1 - \delta_{ij}) \big] \tag{100}$$

with $c = 0.15$, which falls within the range of experimentally observed noise correlations ($c = 0.1 - 0.2$) as reported by Zohary et al. (1994) and Shadlen et al. (1996). These simulations confirmed that our main conclusions remain robust: (1) the network continues to integrate information in a Bayesian manner (see Fig.S3a&b in revised Appendix), and (2) even in the absence of positional information ($I_p = 0$), the network can still eliminate non-local errors caused by grid cell coding, enabling more robust spatial coding (see Fig.S3c in revised Appendix).

### E.2  IMPACT OF COUPLING STRENGTH

We next examine whether variations in coupling strength (denoted as $J_{g_i,p}$ in Eqn. 9) affect the model's ability to integrate information. Specifically, we systematically varied the coupling strength and repeated the experiment

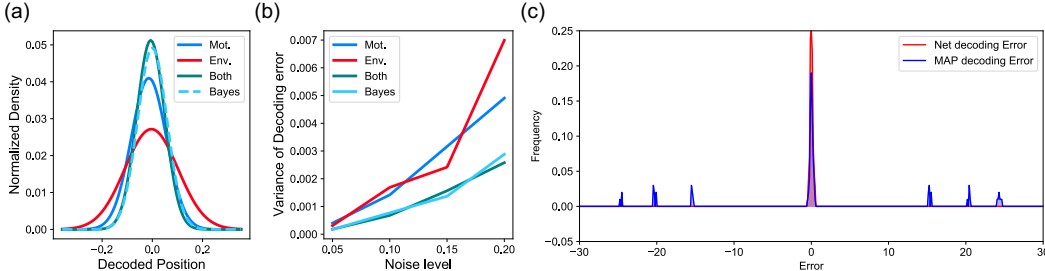

Figure S3: Simulation with Correlated Noise.

shown in Fig. 3 of the main text. The results, presented in Fig. S4, demonstrate a robust alignment between the model's information integration and Bayesian integration across different coupling strengths.

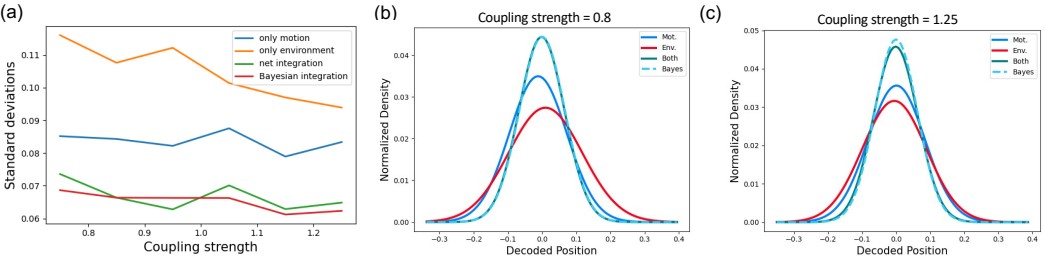

Figure S4: Impact of coupling strength on information integration. (a) Standard deviations of decoding results under different coupling strengths. (b) and (c) Examples of decoding result distributions for varying coupling strengths.

### E.3    MODEL COMPARISON

We compare our model with the "constrained range model" proposed by Sreenivasan & Fiete (2011) (see Fig. S5). Both models are capable of eliminating non-local errors when the decoding range is tightly constrained (Fig. S5a). However, when the decoding range becomes larger, our model continues to eliminate non-local errors, whereas the "constrained range model" fails to do so (Fig. S5b-c).

The key difference lies in how non-local errors are handled. Our model leverages recurrent currents within the place cell network to store historical information, enabling robust error correction without relying on range constraints. In contrast, the "constrained range model" reduces non-local errors by limiting the coding range, thereby sacrificing coding capacity. This distinction highlights the advantage of our model: the dynamics of the place cell network not only ensure robust spatial coding but also preserve coding efficiency.

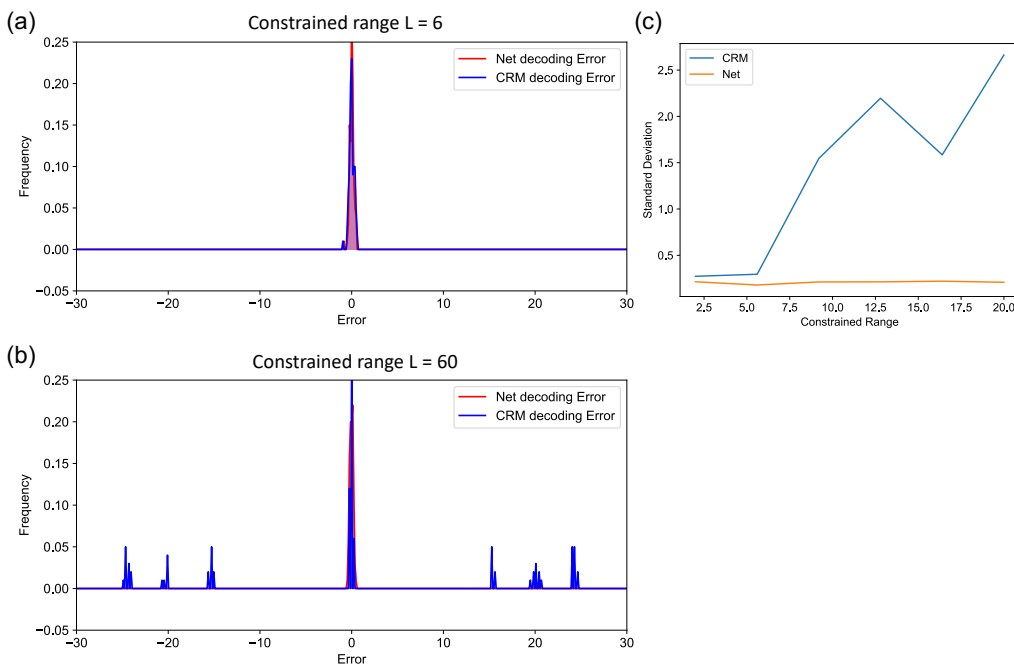

Figure S5: Comparison of our model with the constrained range model. (a) Both models successfully eliminate non-local errors when the decoding range is small. (b) and (c) When the decoding range is large, our model maintains error correction while the constrained range model fails.

