# OpenReview forum: "Complementary Coding of Space with Coupled Place Cells and Grid Cells"
_ICLR.cc/2025/Conference — Submitted to ICLR 2025_

### Official Review · Reviewer_Xuro · 2024-10-28

**Soundness:** 2
**Presentation:** 2
**Contribution:** 2
**Rating:** 5
**Confidence:** 2

**Summary:**

This theoretical paper applies continuous attractor neural networks (CANNs) to model hippocampal place cells (using a single CANN) and entorhinal grid cells (using multiple CANNs). The coupled CANN framework captures correlations between environmental and motion cues, linking place cells in the hippocampus with grid cells in the medial entorhinal cortex (MEC).

**Strengths:**

1. Place cells and grid cells are very important topics in the neuroscience field.

2. This is a strong computational/theoretical paper with 20 equations in the main text and an additional 79 equations in the supplementary text. I briefly went through these equations and did not find any errors.

3. The presented figures are relatively clear.

4. Raw code is uploaded.

**Weaknesses:**

1. I am unsure whether this paper fits ICLR. The primary focus of this study is in the area of "applications to neuroscience & cognitive science." There is no machine learning or deep learning component, only computational neuroscience work. It would likely be a better fit for journals like Neural Computation, PLOS Computational Biology, or Frontiers in Computational Neuroscience, and could also be suited for NeurIPS. I am not aware of ICLR publishing a pure neuroscience paper without a direct ML/DL connection in the past five years, which reflects the primary audience of the conference.


2. There is no validation with experimental data. While theoretical analysis is important and useful, it is not sufficient. Over the past decade, many open-source datasets on place cells and grid cells have become available, such as those from the Buzsáki Lab (https://buzsakilab.com/wp/database/), the Moser Lab (which publishes data with most papers, for example, https://doi.org/10.25493/SKKX-4W3, https://figshare.com/articles/dataset/Toroidal_topology_of_population_activity_in_grid_cells/16764508), and CRCNS (https://crcns.org/data-sets/hc). A recent Nature Neuroscience paper from the Dombeck Lab (2024) also provides raw data on both place cells and grid cells in a virtual linear track (https://www.nature.com/articles/s41593-023-01557-4#data-availability). I think experimental data like this would fit well with Figure 1 in this paper.

**Questions:**

1. Could you please change some of the colors in Figures 3c and 3d? The caption mentions "blue, yellow, and green," but I could not identify any yellow. I don’t believe I am color-blind, so I suggest using the top four colors provided in Matplotlib: ['#1f77b4', '#ff7f0e', '#2ca02c', '#d62728'].

2. In Figure 1d, there is a character "c" under the "$\pi$". Was this intentional?

---

> ### Author Response · Authors · 2024-11-23
> **Response to Reviewer Xuro**
>
> We sincerely thank you for your time and comments on our manuscript. While we acknowledge that some of your concerns pertain to the suitability of our work for ICLR, we hope to clarify its relevance and address the specific points raised. Below, we provide detailed responses to both your major and minor concerns.
>
> **Major concerns:**
>
> 1. Suitability of the manuscript for ICLR:
>
> We appreciate your comments on the alignment of our work with the primary audience of ICLR. However, we respectfully argue that ICLR has increasingly become a platform for computational neuroscience studies. Several recent works focusing on computational modelling of grid cells, rather than direct ML/DL contributions, have been published at ICLR. Examples include:
>
> - Yu et al. (2021): Prediction and generalisation over directed actions by grid cells.
> - Whittington et al. (2021): Relating transformers to models and neural representations of the hippocampal formation.
> - Dorrell et al. (2023): Actionable neural representations: Grid cells from minimal constraints.
> - Whittington et al. (2023): Disentanglement with biological constraints: A theory of functional cell types.
>
> These papers, like ours, do not directly propose new ML algorithms or compare models to experimental data. Instead, they focus on developing theoretical insights into biological systems, advancing the field of computational neuroscience, and will eventually link to brain-inspired intelligence. Given this precedent, we believe our work fits well within the ICLR community’s interest in biologically-inspired approaches to learning and representation.
>
> 2. Experimental validation:
>
> We acknowledge the importance of comparing computational models with experimental data to enhance biological plausibility. However, our current study focuses on developing a theoretical framework for understanding spatial representations in coupled place-grid cell networks. Integrating experimental datasets, such as those from the Buzsáki Lab, Moser Lab, or Dombeck Lab, is an excellent suggestion and could significantly extend this work in the future.
>
> While we do not include a direct quantitative comparison with experimental data in this paper, we have identified qualitative experimental evidence that supports key aspects of our model. For instance, findings from Campbell et al. (2021) align well with our theoretical predictions. Specifically, their results state:
>
> "In V1 and RSC, firing rates peaked 20 cm before each visual landmark (Figures 1I and 1J), with the receptive field location influencing spatial firing rate maps in V1 (Figure S1C). In MEC, the average firing rate was relatively constant over the VR track (Figures 1I and 1J)."
>
> This observation suggests that MEC grid cells may not receive direct landmark cue-based inputs. If such inputs are present, grid cell firing rates would likely vary with proximity to landmarks. In contrast, our model predicts that landmark cues influence grid cells indirectly, mediated through place cells. In our framework, place cells first process landmark cue-based inputs, encoding this information via their attractor dynamics, which inherently stabilize the activity and reduce input variance. This stabilized information is then relayed to grid cells, explaining the lack of landmark proximity effect in their firing rates.
>
> Additionally, Campbell et al. (2021) report that "MEC map shifts were larger when landmark inputs were less certain, although the contrast sensitivity appeared to be sharp." This observation aligns with the Bayesian principle that inputs should be weighted by their uncertainty—a core idea in our model's Bayesian integration of information between place and grid cells.
>
> To enhance the link of our work to experimental data, we will add a discussion of these experimental correspondences in the revised manuscript, demonstrating how our theoretical framework provides a mechanistic explanation of these findings.
>
> **Minor concerns:**
>
> 1.	Figure 3c and 3d colors:
>
> Thank you for pointing out the discrepancy in the figure caption. We will revise the captions of Figures 3 to “blue, red, and green,” ensuring they align with colors in the figure.
>
> 2.	Figure 1d notation:
>
> The “c” under the “π” in Figure 1d was unintended and will be removed in the revised manuscript for clarity.

---

> > ### Comment · Reviewer_Xuro · 2024-11-29
> >
> > Thank you for your comment, and provide the related articles, I don't have further questions. I increase the rate to 5 for this interesting work.

---

### Official Review · Reviewer_xhaK · 2024-10-30

**Soundness:** 3
**Presentation:** 3
**Contribution:** 3
**Rating:** 6
**Confidence:** 3

**Summary:**

The paper presents a computational model that investigates how place cells in the hippocampus (HPC) and grid cells in the medial entorhinal cortex (MEC) collaborate to achieve robust spatial representation. Recognizing that place cells and grid cells have different spatial coding strategies—place cells localize specific positions based on environmental cues, while grid cells encode position through a periodic phase code driven by self-motion—the authors develop a model with reciprocally coupled continuous attractor neural networks (CANNs) to represent these neural populations.

The model, which includes coupled CANNs for place and grid cell networks, demonstrates how reciprocal interactions allow optimal integration of environmental and motion cues, leveraging each system’s strengths with respect to coding while mitigating their respective limitations. The authors theoretically derive that the model effectively performs gradient-based optimization of the posterior distribution of location, achieving Bayesian optimal cue integration. Simulations validate the model’s ability to reduce non-local errors in grid cell phase coding and show that this network configuration leads to accurate spatial representation even with noisy inputs.

The paper’s contributions include:

 - Formulating the interaction between HPC and MEC as a probabilistic inference model for cue integration.
 - Proposing a coupled CANN model to implement this integration and performing gradient-based optimization of the posterior (GOP).
 - Demonstrating through simulations that the model achieves Bayesian optimal integration of sensory cues and mitigates non-local errors in grid coding.
 - This study offers insights into how place and grid cells may interact to enhance spatial coding accuracy and flexibility in the brain, further exploring spatial representation mechanisms addressed in the literature.

**Strengths:**

**Originality:**
The paper presents a novel framework for understanding the complementary roles of place and grid cells by modeling them as coupled continuous attractor neural networks (CANNs). The approach of using reciprocal interactions to enable Bayesian integration of environmental and self-motion cues creatively extends existing ideas on spatial representation. This combination of probabilistic inference with neural dynamics is an innovative contribution to models of hippocampal-entorhinal interaction.

**Quality:**
The theoretical derivation of gradient-based optimization of the posterior (GOP) within the coupled CANN model is rigorous, showing clear mathematical foundations for the proposed integration mechanism. The simulation results are well-designed to validate the model's effectiveness, particularly in achieving Bayesian optimal integration and minimizing non-local errors in grid coding. The use of multiple noise levels to test robustness adds credibility to the results.

**Clarity:**
The paper is generally well-organized, with each section building logically on the previous one. Key concepts, such as the difference between localized space coding (LSC) and phase space coding (PSC), are clearly explained, making the paper accessible even to readers less familiar with neural models of spatial representation. Equations and diagrams are effectively used to support the conceptual flow.

**Significance:**
The model has important implications for understanding how the brain integrates sensory information to form stable spatial maps, a fundamental aspect of cognition and navigation. By addressing the limitations of phase coding in grid cells through a biologically plausible mechanism, the work advances the field's understanding of error correction in neural representations of space. This approach could also inform future work in both neuroscience and neural-inspired AI systems, making the findings broadly relevant.

**Weaknesses:**

1. The model assumes that grid cells receive direct position-based input (eq 13), rather than path-integrated input derived from velocity and head direction signals, as experimental studies suggest. The paper could benefit from incorporating self-motion signals more directly, allowing grid cells to compute position from velocity and direction cues. This also points to potential misalignment with the Gaussian error of eq 13 as path integration would lead to error accumulation and necessitate correction within the grid-place cell network. This error correction is often proposed corrected by border cells [Hardcastle et al.](https://www.sciencedirect.com/science/article/pii/S0896627315002639), and/or place cell inputs [Bonnevie et al.](https://pubmed.ncbi.nlm.nih.gov/23334581/)

2. The model relies on several parameters—like coupling strengths and noise levels—that may significantly impact its dynamics and stability, but these sensitivities are not explored in depth. A systematic analysis of how changes in these parameters affect model performance would strengthen robustness claims and clarify under what conditions the model's optimal cue integration holds. Testing a broader range of parameter values, such as varying the coupling strength between place and grid cells, could also demonstrate how adaptable the model is to different spatial and temporal contexts.

3. The paper validates its model mainly by comparing it to MAP-based decoding, which may not fully illustrate the model’s advantages or limitations compared to other established hippocampal-entorhinal network models. Adding comparisons with additional models, such as attractor networks or the “constrained range” model (Sreenivasan & Fiete, 2011), would provide a more complete evaluation and clarify if this model truly overcomes non-local errors or merely trades one set of limitations for another. Moreover, a comparison with broader literature is missing, such as catastrophic errors [Lenninger et al.](https://elifesciences.org/articles/84531) and integration of landmarks in MEC [Ocko et al.](https://www.pnas.org/doi/10.1073/pnas.1805959115).

4. The claim that the model reduces non-local errors in grid coding is central, but the current simulations only partially support this claim. It would help to design specific tests that induce non-local errors—such as controlled phase shifts or systematically increasing noise levels—to see how effectively the model mitigates them. Additionally, statistical validation of the error reduction compared to baseline models would make this evidence more robust and provide a clearer indication of improvement.

5. The model relies on reciprocal interactions between place and grid cells but does not fully discuss the anatomical evidence supporting this communication. Specifically, mapping the model’s feedback dynamics to known hippocampal projections (e.g., CA1 to MEC layer V or subiculum to MEC layers II/III) could help clarify the model’s relevance to actual neural circuitry. Additionally, exploring the functional implications of these specific pathways within the model could provide a more comprehensive picture of how place and grid cells coordinate spatial representation in the brain.

**Questions:**

The paper is good, but there is insufficient comparisons and alignment to existing literature, to change my opinion I would like to see the weaknesses addressed as well as

## Comparison with models
Given that the paper uses Continuous Attractor Neural Networks (CANNs) to model place and grid cell interactions, specific comparisons with alternative models should focus on how well CANNs address spatial stability, error correction, and flexibility in comparison to other neural network models that represent spatial information, particularly in grid and place cells.

### 1. Comparison with Classic Attractor Networks
   - While CANNs represent continuous variables well, they can suffer from drift and boundary effects. Comparing this model to classic attractor networks (like ring or torus attractors) could reveal whether CANNs offer better stability over large spatial maps. Moreover, there are several models with integration of landmarks in CANNs that should be discussed [Ocko et al.](https://www.pnas.org/doi/10.1073/pnas.1805959115), [Campbell et al.](https://pmc.ncbi.nlm.nih.gov/articles/PMC6205817/)
   - Classic attractors often use recurrent feedback to “snap” activity patterns back into place when noise occurs. A direct comparison would show if the CANN model is more robust or if it encounters similar drift issues.

### 2. Comparison with the Constrained Range Model (Sreenivasan & Fiete, 2011)
   - The constrained range model uses non-overlapping spatial scales to extend range without aliasing. Comparing this to the CANN model could clarify if CANNs handle large, unambiguous spatial ranges better.
   - The constrained range model achieves wide spatial coverage with few grid scales. Evaluating whether the CANN model requires more parameters or fine-tuning to achieve similar coverage would reveal if it offers any unique advantages in precision or robustness.

## Comparison with experimental literature
In the paper, you claim that place cells and grid cells recieve independent input, which are cue-based and self-motion-based, respectively. However, this does not coincide with recent literature on cue integration in grid cells; please justify your claims wrt [Campbell et al. 2018](https://pmc.ncbi.nlm.nih.gov/articles/PMC6205817/), [Campbell et al. 2021](https://pubmed.ncbi.nlm.nih.gov/34496249/). Of particular interest is the latter paper, which concludes, "Our gain change experiments in low-contrast conditions revealed that MEC map shifts were larger when landmark inputs were less certain, although the contrast sensitivity appeared to be sharp. This finding is consistent with the general Bayesian principle that inputs should be weighted according to their certainty". These recent results could be a very nice opportunity to align model predictions with experimental findings.

**Primes and approximations**
While primes provide an elegant theoretical tool for modeling large, unambiguous spatial ranges, there is no direct evidence that grid cells employ prime numbers for encoding spatial information. The observed ratios in experimental studies are not exact primes; they are approximately constant ratios, around 1.4–1.7 between adjacent modules [Stensola et al.](https://www.nature.com/articles/nature11649)

Moreover, the approximation $\prod_i^M \lambda_i\approx \bar{\lambda}^M$ is not justified when $\lambda_i$´s are primes, so why use this approximation when $\frac{\ln\left(\prod_{i=1}^M \lambda_i\right)}{M N_0} = \frac{\overline{\ln \lambda}}{N_0}$ gives the same conclusion?

---

> ### Author Response · Authors · 2024-11-22
> **Reviewer xhaK**
>
> We sincerely thank Reviewer xhaK for their thoughtful and detailed feedback. Below, we address each of the concerns raised, providing explanations, additional analyses, and revisions where necessary.
>
> 1. About the position-based, rather than motion-based, input to grid cells in our model:
>
> Thanks for the thoughtful suggestion. In the present study, our primary focus is not on how grid cells perform path integration, but rather on how the coupled networks between place cells and grid cells efficiently integrate information from different modalities and they eliminate non-local errors in the absence of external environmental cues. In our model, the position-based inputs received by  grid cells may represent signals  from upstream cells performing path integration. These inputs could originate from other grid cells, as suggested in path integration models of grid cells (e.g., Giocomo et al., 2011), or potentially from other cell types, such as band cells (e.g., Krupic et al., 2012).
>
> While we acknowledge that directly incorporating self-motion signals into the model could provide a more comprehensive understanding of grid cell dynamics, as a first step, which is already a novel contribution in the field, the present study will only focus  on how the interactions between place cells and grid cells enhance spatial coding robustness, especially under conditions where external cues are unavailable. We will certainly consider your  insightful feedback in future work.
>
> 2. About parameter sensitivity of the model:
>
> Thank you for raising these important issues. We have done some of them in the current manuscript. For example,  in Fig. 3d, we  compared the network's information integration with Bayesian integration under varying noise levels. As suggested by the reviewer,
> to further validate the robustness of our model, we have conducted new experiments by varying the coupling strength between place and grid cells. These new results, now included in the appendix of the revised manuscript, show that the network performs near-optimal Bayesian information integration across a range of coupling strengths (see Fig. S4).
>
> 3. About demonstration of non-local error elimination of the model:
>
> Thanks for the suggestions. We would like to clarify that some of them have  been done in the current manuscript. As shown in Fig. 4a, we compared the error distributions of the coupled network and the MAP (baseline model) under an identical noise condition. Furthermore, as shown in Fig. 4b, we systematically varied the input noise strength and compared how the error variances of both decoding methods change with the increasing noise level. These results demonstrate that the coupled network significantly reduces non-local errors compared to MAP,  supporting  the statement that our model effectively mitigates non-local errors.
> As suggested by the reviewer, we will perform a statistical p-test to quantitatively assess the differences between the network decoding results and the baseline model presented in Fig. 4a.  This analysis will be included in the revised manuscript.
>
> 4. About discussion about mapping the model’s feedback dynamics to known hippocampal projections:
>
> Thanks you for the suggestion. In the current  model, we consider that place cells and grid cells are directly connected. This simplified model gives us insight into understanding how place and grid cells interact with each other to improve spatial representation. We recognize that in the actual neural system, the projections from grid cells to place cells are via direct inputs from MEC Layer II/III to CA3 and CA1, and the projections from place cells to grid cells involve indirect pathways—such as projections from CA1 to MEC Layer V or the subiculum, followed by internal MEC connections back to Layer II/III. We also acknowledge that incorporating more biologically detailed structures will allow our  model to better align with the anatomical evidence. In future work, we plan to extend the current  model to include these specific pathways and explore their implications for spatial representation. Nevertheless,
> in the revised manuscript, we will add a discussion of these anatomical pathways and their potential influences on our results.

---

> ### Author Response · Authors · 2024-11-22
> **Follow-up response**
>
> 5. Comparison with existing models:
>
> Thanks for the detailed comments and insightful suggestions. We appreciate the opportunity to clarify and expand on the comparisons between our model and the existing literature.
>
> •	First, regarding comparisons with classic attractor networks. Actually, our coupled network belongs to classic attractor models, such as we used 1D CANN (i.e., line attractor network) for place cells and multiple ring attractors for grid cells.  In the revised manuscript, we will include discussion of models that incorporate landmarks into attractor networks, such as those proposed by Ocko et al. and Campbell et al.. Unlike these models, which rely on external environmental cues (e.g., landmarks) to correct path integration errors, our work focuses on how place cells contribute to improving grid cell coding in the absence of external information—such as during navigation in darkness. Specifically, in our model, error correction is achieved through the storage of historical  information in the attractor dynamics of the place cell network.
>
> •	Second, following the reviewer’s suggestion, we have added a direct comparison with the "constrained range model" proposed by Sreenivasan & Fiete (2011) in Appendix of the revised manuscript (Fig. S5). The results show that both models can eliminate non-local errors under small, constrained decoding ranges. However, when the decoding range increases, our model continues to perform robustly, while the constrained range model fails. This distinction arises from that our model leverages the recurrent dynamics of the place cell network to store the history information, enabling error correction without sacrificing spatial coding capacity. In contrast, the constrained range model mitigates non-local errors by limiting the spatial range, which inherently sacrifices the coding capacity.
>
> •	Regarding the total coding range of grid cells, both our model and the model proposed by Sreenivasan & Fiete employ phase combination coding, resulting in equivalent coding capacity theoretically. However, in practice, since the constrained range model requires decoding within a small, constrained range to avoid non-local errors, its true coding capacity becomes very limited; whereas, our model always achieves large effective coding range as it relies on the coupling between grid and place cells.

---

> ### Author Response · Authors · 2024-11-22
> **Follow-up responses**
>
> 6. Comparison with experimental literature:
>
> Thanks for pointing out these valuable references. We have read the studies by Campbell et al. (2018, 2021) and agree that they provide an excellent opportunity to align our model predictions with recent experimental findings, enhancing the connection between our theoretical framework and real data.
>
> First, we would like to address the reviewer’s concern regarding our claim that place cells and grid cells receive independent inputs (cue-based and self-motion-based, respectively). In our model, while grid cells and place cells receive independent self-motion and environmental cues, respectively, they are connected reciprocally. This implies that the activity of grid cells is influenced by the environmental cue via the connection from  place cells. In other words, the dependence of grid cells’ activity on the environment cue does not mean grid cells receive the environmental cue directly.
>
> Actually,  the findings in Campbell et al. (2021) tend to support  our model. For example, the results section of their paper states:
> "In V1 and RSC, firing rates peaked 20 cm before each visual landmark (Figures 1I and 1J), with the receptive field location influencing spatial firing rate maps in V1 (Figure S1C). In MEC, the average firing rate was relatively constant over the VR track (Figures 1I and 1J)."
> This observation suggests that MEC grid cells do not receive direct landmark cue-based inputs. If they do, we would expect grid cells’ firing rates  vary with proximity to landmarks. In contrast, our model predicts that landmark cue reaches grid cells indirectly, mediated through place cells. In our framework, place cells first receive the landmark cue-based input and then encode this information via the attractor dynamics, which inherently reduces the input variance by stabilizing the activity into an attractor state. This filtered information is then relayed to grid cells, which justifies why grid cells’ firing rates remain unaffected by landmark proximity.
>
> Moreover, Campbell et al. (2021) reported that "MEC map shifts were larger when landmark inputs were less certain, although the contrast sensitivity appeared to be sharp.". This finding aligns well with the Bayesian principle that inputs should be weighted according to their uncertainty—a fundamental idea underlying our model’s Bayesian integration of information between grid cells and place cells. In this sense, our model provides a theoretical explanation for these experimental observations.
>
> In the revised manuscript,
> we will incorporate a discussion of these findings to highlight how the model’s predictions align with experimental results. In future work, we  will quantitatively compare our model predictions with the experimental data from Campbell et al. (2021).

---

> > ### Comment · Reviewer_xhaK · 2024-11-26
> >
> > Thank you for your detailed response and added results. Happy that you found the comments helpful.
> >
> > I have no further questions or comments.

---

> > > ### Author Response · Authors · 2024-12-02
> > > **Thank You for Your Feedback**
> > >
> > > Dear Reviewer,
> > >
> > > Thank you for taking the time to review our response and for your thoughtful comments throughout the review process. We are glad that you found our responses and the additional results helpful.
> > >
> > > We greatly appreciate your engagement with our work and your valuable feedback, which has significantly contributed to improving the clarity and rigor of our manuscript. We sincerely hope that our efforts have addressed your concerns and that you might consider raising your score for our submission.
> > >
> > > Thank you again for your time and consideration.

---

### Official Review · Reviewer_iXH2 · 2024-11-03

**Soundness:** 1
**Presentation:** 1
**Contribution:** 2
**Rating:** 5
**Confidence:** 3

**Summary:**

This paper constructs a neural network model of recurrently coupled grid and place cells which performs maximum a posteriori (MAP) estimation of the animal's position. The model is somewhat biologically plausible and is surprisingly tractable to theoretical analysis, despite its relative complexity.

**Strengths:**

The paper is somewhat slow to get to the main point and results in section 4 + 5. Once it gets there, the basic analysis and numerical experiments are pretty interesting and intuitive. In essence, the paper shows that the place cell attractor network can select the most likely position through what is essentially winner-take-all dynamics.

This idea is intuitive, but the authors do a reasonably good job of formalizing this intuition. The theoretical analysis is *very* detailed &mdash; in fact, probably too many details are provided in the main text. I would like to see section 5 expanded and section 4 stripped down to the most fundamental equations (e.g. relegate equations 12+13 to the appendix).

**Weaknesses:**

* The analysis assumes that neural response noise is independent across all neurons, a common assumption which is well-known to be violated in real biological systems (e.g. Abbott & Dayan, 1999). This limitation to the analysis is not discussed by the authors in detail.

* The model makes various parametric assumptions about the nature of spatial tuning and it is unclear how sensitive the analysis is to these assumptions. For example...
    * Equation 1 posits a unimodal Gaussian tuning curve for place cells &mdash; not only are place cells not really Gaussian, but they are often multi-modal in large environments (e.g. [Fenton et al 2008](https://www.jneurosci.org/content/28/44/11250.short) and [Rich et al. 2014](https://doi.org/10.1126/science.1255635)).
    * Equation 5 posits a Gaussian noise model which is also biologically unrealistic, relative to other common choices, e.g. Poisson.
    * The different grid cell modules are not reciprocally connected, which seems like a dubious assumption.
    * Overall, it is not clear to me that these parametric assumptions are critical to the main story that the authors make. If they are not critical, this should be demonstrated directly. Furthermore, if they are not critical to the main results, then many of these equations in sections 3 and 4 should be relegated to the Appendix/Supplement as they are highly distracting.

* The authors claim that "optimal decoding is achieved by maximum a posterior (MAP)" (line 220). At best this is a mathematically incomplete statement. At worst, it's flatly incorrect. The optimal decoding rule will depend on the loss function you posit on your point estimator. The authors do not appear to formalize a loss function on the decoder, but the most common choice is to use the expected squared error. Under this choice, the *posterior mean and **not** the posterior maximum/mode* is the optimal point estimate. My complaint here is pedantic and fixable, but on the other hand this is very basic stuff (see "Examples" section on [the wikipedia page for Bayes optimal estimators](https://en.wikipedia.org/wiki/Bayes_estimator)) and it doesn't inspire my confidence when papers miss details like this that are central to their narrative.

* The critical theoretical prediction of the paper seems to be equation 19, which shows that the dynamics of the network perform gradient ascent on the posterior. It would be great to include a numerical simulation showing the accuracy of this prediction (since it relies on a few simplifying assumptions).

* More citations and references should be provided throughout the manuscript. For example, in section 2 the work of Fiete et al (2008) should be cited when explaining the importance of having co-prime factors, and in section 2.2 there should be a reference about Fisher information and Cramer Rao bound (which is alluded to but not explicitly mentioned). Also, MacKay's textbook is cited in two different formats in the references (once as MacKay David 2022 and another time as David JC MacKay 2003). The citation to MacKay's book at the beginning of section 2.1 (here it is cited inline as David 2022) is puzzling to me. Which chapter / section are you referring to?

* The main point of the paper seems to be that one can construct a recurrent neural network to de-noise an estimate of an external variable (via MAP inference in this case). This conceptual point doesn't actually strike me as that novel &mdash; it is the basic idea behind Hopfield networks. Bayesian interpretations of Hopfield networks (e.g. in [this paper](https://doi.org/10.1109/ICNN.1993.298580)) and of continuous attractor networks (e.g. in [this paper](https://doi.org/10.1073/pnas.2210622120)) have been put forth in the literature before and don't seem to be cited / discussed by the authors.

* I hate to say it, but I think this paper is likely a better fit for a physics journal or a neuroscience journal than ICLR. There is very minimal appeal to this flavor of work to the broader ML / AI community. Overall, I'm okay with including papers like this in ICLR, but the area chair may feel differently.

**Questions:**

None

---

> ### Author Response · Authors · 2024-11-22
> **Response to Reviewer iXH2:**
>
> We sincerely thank Reviewer iXH2 for their thoughtful and detailed feedback, which has greatly helped us identify areas to refine and improve our manuscript. Below, we address each of the concerns raised, providing explanations, additional analyses, and revisions where necessary.
>
> Weakness 1:
> - Noise dependence across neurons.
>
> Response:
> - Thank you for raising pointing this issueout. Indeed, as discussed in the study by Abbott & Dayan (1999), neural noise is not independent across neurons in experimental results obtained from in vitro slice preparations. In our work, we assumed independent noise to simplify the network dynamics, allowing for a concise correspondence with the gradient-based optimization of the posterior (as shown in Eqs. (17) and (18) in the main text). However, including neural correlation does not change our main conclusions. To confirm this, we conducthave also conducted simulations includingtroducing correlated noise in the network dynamics, where the noise covariance is modeled as,
> $$〈(r_i-f_i )(r_j-f_j )⟩=\sigma^2 [\delta_ij+c(1-\delta_ij )] $$
> with $c=0.15$, which falls within the range of experimentally experimentally observed noise correlations ($c\in(0.1,0.2)$) as reported by Zohary et al. (1994) and Shadlen et al. (1996). These simulations confirmed that our main conclusions remain unchangedrobust: (1) the network continues to integrate information in a Bayesian manner (see new Fig.S2 a&b in the revised Appendix), and (2) in the absence of positional information ( $I_p=0$ ), the network can still eliminate non-local errors caused by grid cell coding, enabling more robust spatial coding (see new Fig.S2 c in the revised Appendix).
>
> Weakness 2:
> - Parametric assumptions of the model.
>
> Response:
> - Thank you for raising these important issues regarding  parameter assumptions in our model. We address them one by one below:
> 1.	Unimodal Gaussian tuning curve for place cells (Equation 1):
> By definition, place cells are neurons that exhibit localized place fields in an environment. While it is true that some hippocampal neurons show multiple place fields in large environments (e.g., Fenton et al., 2008; Rich et al., 2014), a significant proportion of place cells have single, localized place fields (e.g., see Fig. 4b in Rich et al., Science, 2014). Additionally, neurons exhibiting multiple fields may result from remapping, where the animal perceives a large environment as multiple sub-environments. In our model, the Gaussian tuning curve is a mathematical approximation to facilitate theoretical tractability. We believe that alternative bell-shaped tuning curves (e.g., sigmoidal or cosine tuning) would yield similar results and do not qualitatively affect the model's conclusions, as confirmed by many previous modelling studies.
> 2.	Gaussian noise model (Equation 5):
> In our firing-rate model, each unit represents a population of neurons encoding the same position (e.g., place cells) or sharing the same phase (e.g., grid cells). While individual neurons typically exhibit Poisson spiking behavior, the firing rates averaged across a population approximate a Gaussian distribution due to the central limit theorem. This mean-field approach is widely used in modeling study and provides a biologically reasonable simplification. Thus, assuming Gaussian noise is consistent with the aggregated firing rate behavior of neuronal populations and does not diminish the validity of our results.
> 3.	Independent grid cell modules:
> Experimental evidence suggests that grid cell modules are anatomically separated, with very weak interconnectivity (Hafting et al. 2005). Therefore, studies such as Sreenivasan and Fiete (2011) and Agmon and Burak (2020) treated different grid cell modules as independent in their theoretical models. Additionally, introducing weak random connections between grid cell modules won’t significantly affect our simulation results.
> Overall, we  would like to point out that these parametric assumptions are common in theoretical modelling studies and they are not critical to the core conclusions of our work. They mainly contribute to   facilitating a tractable theoretical framework for analyzing the coupled dynamics of place cells and grid cells.
> We hope that our responses and revisions have addressed  the concerns of the reviewer. Thanks again for the valuable comments.

---

> ### Author Response · Authors · 2024-11-22
> **Follow-up responses**
>
> Weakness 3:
> - Loss function of "Optimal Decoding"
>
> Response:
> - Thanks for bringing this issue to our attention. We have revised the manuscript to clarify the relationship between the MAP estimator and its associated loss function. Specifically, in lines 220–221 of the revised main text, the original statement "Mathematically, optimal decoding is achieved by maximum a posterior (MAP)" has been modified to: "Maximum a posteriori (MAP) decoding is a widely used method (references) and proven to be the optimal decoding strategy under a 0-1 loss function." This revision provides a more accurate and complete explanation of our approach. We sincerely appreciate your feedback on this important detail.
>
> Weakness 4:
> - Direct comparison between network dynamics and Gradient ascent of posterior
>
> Response:
> - Thanks for this suggestion. In Fig. 3b of our paper, we compared the decoding results obtained from the network with those obtained via gradient-based optimization of the posterior (GOP). But as you pointed out, we did not explicitly include the trajectory of the network activity during the decoding process (before reaching the steady state) and compare it to the trajectory of the gradient ascent-based optimization. We have now added examples illustrating the network trajectory and its alignment with the gradient ascent trajectory. These results will be added to the revised Appendix for completeness (see the new Fig. S2).
>
> Weakness 5:
> - More citations and references
>
> Response:
> - Thank for pointing out these issues. We have addressed them in the revised manuscript. Specifically:
> 1. A citation to Fiete et al. (2008) was added in section 2 to explain the importance of co-prime factors (see line 154).
> 2. A reference to the Cramér–Rao bound was included in section 2.2 (lines 162–163): "We use Fisher information, whose inverse is the lower bound of decoding variance of any unbiased estimator (commonly known as the Cramér–Rao bound (Cramér, 1946))."
> 3. The citations to MacKay's textbook were standardized throughout the manuscript to ensure consistency, and the reference has been updated to indicate that we are specifically referring to Chapter 3.
>
> We appreciate your feedbacks, which  help us to improve the clarity and accuracy of the manuscript.
>
>
> Weakness 6:
> - Discussion about other works of Bayesian intepretation of attractor networks
>
> Response:
> - Thank you for pointing out these two relevant studies, both of which discussed the relationship between attractor networks and Bayesian inference. These works share some common ground with our study, and we will add a paragraph discussing them in the revised manuscript. However, there are significant differences between our work and these studies:
> 1. Our model is not a single-layer attractor neural network, rather   it consists of  a line attractor network formed by place cells,  which is reciprocally coupled with multiple ring attractor networks formed by grid cells
> 2. The focus of our paper is not on proposing a  Bayesian interpretation of attractor dynamics. Instead, we investigate: 1) how the interactions between the place cells’ network and the grid cells’ networks enable the efficient integration of multimodal information, where the reciprocal connections between place and grid cells convey the correlation prior; 2),  how the coupling with  place cells help grid cells to resolve the non-local error problem.
>
> To our best knowledge, our results are novel in the study of how place cells and grid cells interact with each other to improve spatial representation in the brain, and they are valuable contributions to the field.
> We hope that our replies have addressed all concerns of the reviewer and could persuade the reviewer to raise the score.

---

> > ### Comment · Reviewer_iXH2 · 2024-11-25
> > **Reviewer Response**
> >
> > I thank the authors for their detailed and thoughtful response. After careful consideration and reading the other reviewer comments, I feel comfortable with retaining my score at a 5 (slightly below acceptance threshold).
> >
> > Most critical in my mind is weakness #6, which questions whether there is a strong conceptual advance here over existing modeling work. This weakness is fixable but would require significant revisions that I think go beyond the scope of what can be done in an ICLR rebuttal period. I do think that this paper is a better fit for a venue like PLOS computational biology or Neural Computation, where reviewers will ask for more detailed discussion and comparisons of this model to prior work. If I were reviewing this for one of these journals my recommendation to the editor would be for a "revise and resubmit", but unfortunately the revisions are currently too substantial in my opinion for the ICLR process.

---

> > > ### Author Response · Authors · 2024-12-02
> > >
> > > Dear Reviewer,
> > >
> > > Thank you for your thoughtful consideration and detailed comments on our response. We sincerely appreciate the time you have taken to evaluate our work and provide constructive feedback.
> > >
> > > We would like to respectfully address your main concern regarding the conceptual advance of our work in comparison to existing models, as raised in Weakness #6. While we understand your perspective, we would like to emphasize the significant differences between our study and the existing works on attractor networks:
> > >
> > > Model Differences: Our framework is not a single attractor network but consists of multiple coupled continuous attractor neural networks (CANNs). Specifically, our model includes a line attractor network representing place cells, which is reciprocally coupled with multiple ring attractor networks representing grid cells. This structure allows us to explore interactions between these systems in novel ways that go beyond previous single-layer attractor network models.
> > >
> > > Conceptual Advances: Our work does not merely offer a Bayesian interpretation of attractor network dynamics. Instead, we investigate how interactions between place cells and grid cells enable the removal of non-local errors in grid cell coding. This mechanism improves the robustness of spatial representations, a key problem in the field that has not been addressed in prior work. Our findings provide new insights into how multimodal information is integrated and how spatial coding becomes more robust in neural systems.
> > >
> > > We hope these distinctions clarify the novelty and significance of our contributions and address the concern about overlap with existing models. We sincerely believe that these advances merit further consideration and respectfully request that you reconsider your score in light of these points.
> > >
> > > Thank you again for your thoughtful review and for contributing to the improvement of our work.
> > >
> > > Best regards

---

### Meta-Review · Area_Chair_s4ja · 2024-12-20

**Metareview:**

The computational neuroscience paper describes and analyzes a model of a coupled network of place cells and grid cells to explain how these complementary mechanisms lead to optimality in the context of probabilistic inference for cue integration. The reviewers appreciated the formulation and the rigorous mathematical analysis of the coupled network model. However, this paper still only received borderline reviews. Reviewers cited concerns about insufficient novelty compared to previous work on attractor networks in the context of Bayesian inference. There was some consensus that perhaps this work could benefit from a more thorough review process facilitated by a computational neuroscience/physics/computational biology journal, as well as some thought that perhaps such a venue might yield a better audience than the ICLR community for this theoretical neuroscience work.

**Additional Comments On Reviewer Discussion:**

There was some back and forth with the reviewers during the discussion, however it appeared like the reviewers would have preferred a more neuroscience/physics journal-like process for this particular paper.

---

### Decision · Program_Chairs · 2025-01-22

Reject